# Examining how goals of care communication are conducted between doctors and patients with severe acute illness in hospital settings: A realist systematic review

Jamie Gross[1,2], Jonathan Koffman[3]*

1 Northwick Park and Central Middlesex Hospitals, London North West University Healthcare NHS Trust, Harrow, United Kingdom, 2 King's College London, Cicely Saunders Institute, Florence Nightingale Faculty of Nursing, Midwifery & Palliative Care, London, United Kingdom, 3 Hull York Medical School, Wolfson Palliative Care Research Centre, University of Hull, Hull, United Kingdom

* Jonathan.koffman@hyms.ac.uk

**Data Availability Statement:** All relevant data are within the paper.

**Funding:** The authors received no specific funding for this work.

## Abstract

### Background

Patient involvement in goals of care decision-making has shown to enhance satisfaction, affective-cognitive outcomes, allocative efficiency, and reduce unwarranted clinical variation. However, the involvement of patients in goals of care planning within hospitals remains limited, particularly where mismatches in shared understanding between doctors and patients are present.

### Aim

To identify and critically examine factors influencing goals of care conversations between doctors and patients during acute hospital illness.

### Design

Realist systematic review following the RAMESES standards. A protocol has been published in PROSPERO (CRD42021297410). The review utilised realist synthesis methodology, including a scoping literature search to generate initial theories, theory refinement through stakeholder consultation, and a systematic literature search to support program theory.

### Data sources

Data were collected from Medline, PubMed, Embase, CINAHL, PsychINFO, Scopus databases (1946 to 14 July 2023), citation tracking, and Google Scholar. Open-Grey was utilized to identify relevant grey literature. Studies were selected based on relevance and rigor to support theory development.

### Results

Our analysis included 52 papers, supporting seven context-mechanism-output (CMO) hypotheses. Findings suggest that shared doctor-patient understanding relies on doctors

**Competing interests:** The authors have declared that no competing interests exist.

being confident, competent, and personable to foster trusting relationships with patients. Low doctor confidence often leads to avoidance of discussions. Moreover, information provided to patients is often inconsistent, biased, procedure-focused, and lacks personalisation. Acute illness, medical jargon, poor health literacy, and high emotional states further hinder patient understanding.

## Conclusions

Goals of care conversations in hospitals are nuanced and often suboptimal. To improve patient experiences and outcome of care interventions should be personalised and tailored to individual needs, emphasizing effective communication and trusting relationships among patients, families, doctors, and healthcare teams. Inclusion of caregivers and acknowledgment at the service level are crucial for achieving desired outcomes. Implications for policy, research, and clinical practice, including further training and skills development for doctors, are discussed.

## Introduction

Involving patients in medical decisions improves patient satisfaction [1, 2], affective-cognitive outcomes[3], allocative efficiency [4] and reduces unwarranted clinical variation [5]. The operational definition of goals of care states "the overarching aims of medical care for a patient that are informed by patients' underlying values and priorities, established within the existing clinical context, and used to guide decisions about the use of or limitation(s) on specific medical interventions" [6]. Its importance relates to the promotion of patient autonomy and patient-centred care, the promotion of valued but avoidance of unwanted care, and the psychological and emotional support provided to patients and families at a time of increased vulnerability [6].

However, in the acute hospital setting true patient-centred care may be more nuanced [7]. Inter-personal and professional differences in knowledge, values, relationships and trust within the confines of a complex healthcare system have witnessed low patient and family involvement in goals of care planning with their doctor [8–17]. In crises, for example in worsening critical illness such as progressive septicaemia, this may be further exacerbated by fluctuant states in the mental capacity of patients from acute illness. High emotional states and other external influences, including lack of time, necessitate decisive action by doctors and can also hinder goals of care discussions taking place between doctors and patients and their families [9, 18–23].

Current evidence supports the reality that patient involvement in goals of care decision-making in hospitals is sub-optimal. Le Guen and colleagues demonstrated that only 12.7% of elderly attendees (aged ≥ 80 years) presenting at an emergency department with a condition potentially requiring intensive care were consulted. A United Kingdom (UK) ethnographic study also demonstrated a low level of engagement with patients and families for Intensive Care Unit (ICU) admission decisions [24]. Reasons included the inability of doctors and patients and/or their families to meet at mutually convenient times and the impact of acute illness on some patients' ability to have meaningful conversations. This qualitative work informed an ICU decision-making model that included patients' wishes and values. However, it does not inform health providers of the practicalities of achieving this, nor does it recommend the desired nature of relevant doctor-patient-family interactions.

A UK nationwide quality improvement initiative—the Recommended Summary Plan for Emergency Care and Treatment (ReSPECT) programme—aims to encourage healthcare professionals and patients to discuss and co-plan for emergency care [25]. However, its widespread adoption into clinical practice has been variable. Subsequent follow-up evaluation studies reveal mismatches between doctors' clinical priorities, the immediate needs of patients and families and shared understanding [22, 26].

Optimal practice should strive to promote meaningful and trusted conversations between doctor and patient about goals of care so that best-interest decisions can be made that are informed and understood by both patient and doctor. This can only be achieved when patients have the appropriate understanding of their illness, the treatment options available, prognosis and degree of uncertainty relating to these, and doctors have an understanding of patients' cultural and personal values and the skill to incorporate these into any decision-making process.

These studies suggest that there is an incomplete understanding of the key drivers and barriers to initiating goals of care conversations and what the most effective communication approaches are between healthcare professionals and patients. Little is also known about the external influences and biases in information exchange.

This study therefore aims to identify and critically examine factors that influence: i) the extent to which goals of care conversations occur between doctor and patient in acute illness in acute hospital settings and ii) how these goals of care conversations are conducted in acute illness.

## Methods

The UK Medical Research Council guidance on the development and evaluation of complex interventions [27] and the Methods of Researching End-of-Life Care statement [28] recommends that new healthcare-related interventions are most likely to be effective when they are underpinned by a "conceptual framework"[28]. This includes a theoretical understanding of the key processes involved in delivering interventions and the contexts in which they are required to operate. This realist review specifically addresses the requirement for theory and conceptual framework development and was developed in December 2021 and published in PROSPERO (registration number: CRD42-21297410 (https://tinyurl.com/mpwsubx4). Realist reviews are a theory-driven systematic approach that is particularly suited to helping understand causation; they aim to investigate what works (or fails to work) for whom, in what circumstances, and how, by identifying processes (mechanisms) that lead to desired outcomes in particular contexts [29, 30]. Furthermore, they examine how mechanisms or 'underlying causal forces or powers' are triggered in particular contexts and lead to outcomes [30]. This specifically relies on using 'context-mechanism-output' configurations (CMOs); these represent testable hypotheses that explain how the context can trigger mechanisms and lead to a variety of outcomes [30]. A three-staged approach was adopted using the RAMSES realist standards [30] and Pawson's realist methodology [29].

### 1. Defining the scope of review by concept mining and theory building

A scoping literature search was performed initially between 1 December 2021 to 24 February 2022 to clarify the purpose of this review and generate initial theories. The PubMed database and Google scholar were primarily used to seek articles relating to the study's aims. These included systematic and non-systematic review articles, key primary studies and any other article type that had relevance to goals of care discussions in acute hospital illness. Knowledge acquisition and abductive reasoning informed the construction of seven preliminary

**Table 1. Context-mechanism-output configuration to generate hypotheses for realist synthesis.**

| Term | Explanation |
|---|---|
| Context | Pre-existing structures, settings, environments, circumstances or conditions that shape whether certain behavioural and emotional responses (for example mechanisms) are subsequently triggered. |
| Context-mechanism-outcome configurations (CMOs) | Describe the causal relationships between contexts, mechanisms and outcomes, that is, how certain outcomes are realised through mechanisms that are triggered in certain circumstances and contexts. |
| Mechanisms | The behaviour or emotional response that is triggered in certain contexts. The mechanism is context-specific and is usually hidden. |
| Outcomes | The final impact of mechanisms that are triggered in certain contexts. |

hypotheses, constructed in a Context-Mechanism-Output (CMO) format (i.e. "if (C). . .then (M). . .which results in (O)" (Table 1).

Adapted from Mitchell et al [31], Papoutsi et al [32] and Cottrell et al [33]

## 2. Stakeholder consultation and refinement of initial theory

The CMO hypotheses were refined following consultation with 16 key stakeholders that had clinical, research or patient experience relating to goals of care conversations and decision-making, an approach adopted in previous realist reviews [33, 34].

Stakeholders were identified within existing networks of both authors and selected based on their personal and/or professional background, their level of experience and if it was felt that they would add value to theory building. Efforts were made to involve a range of different stakeholders from different personal and professional backgrounds to give a rounded insight. Discussions were held individually and were face-to-face, via video or audio call depending on their preference. The 16 stakeholders consulted included 10 hospital-based doctors (specialising in intensive care medicine, general medicine, emergency medicine and palliative care), one specialist nurse (critical care outreach), two hospital managers, one physiotherapist and one former patient with experience of having been cared for in an ICU, and their spouse. Each consultation lasted between 30–60 minutes. Each interview involved a short presentation of the latest iteration of the CMO hypotheses and open-ended questions to ascertain their general thoughts and feelings towards these and whether or not they would recommend any modifications based on their experiences. Field notes were taken with reflective summaries and modifications made to the CMOs where appropriate. When there was doubt and/or ambiguity, agreement was sought between JG and JK. Table 2 presents the resulting CMO following consultation with all 16 stakeholders with relevant modifications made.

## 3. Searching for and appraising the evidence

The refined CMO hypotheses informed a search strategy for the main literature search (S1 Appendix). Searches were run using Medline, Pubmed, Embase, CINAHL, PsychINFO and Scopus databases from 1946 to 14 July 2023. Purposive searching for additional relevant articles that would contribute to programme theory building and theory testing was performed iteratively. These were identified by citation tracking from papers already identified and Google Scholar. Relevant non-published grey literature was also sought using OpenGrey and Google search engines. The screening and selection of articles were based on (i) relevance: whether they contribute to theory building and testing and (ii) rigour: whether the method used to generate that particular piece of data is credible and trustworthy [35]. Although there were no geographic restrictions, papers were limited to those written in English.

**Table 2. Proposed CMOs following scoping literature search and stakeholder engagement.**

| Context (if. . .) | Mechanism (then. . .) | Outcome (which will. . .) |
|---|---|---|
| 1. If patients are provided, understand, and accept information that is delivered in a personalised way about the nature of their illness, the benefits and burdens of life-sustaining treatments and potential outcomes | then they will be more informed | Which will more likely allow a shared understanding between doctor and patient relating to treatment goals and priorities in severe acute illness |
| 2. If the information provided to patients about the benefits and burdens of life-sustaining and non-life-sustaining treatments and potential outcomes is influenced by cognitive bias and other external factors | then it will affect the judgement patients make about their wishes for future care | Which will threaten the likelihood of a shared decision-making approach between doctor and patient towards severe acute illness. |
| 3. If there is a mutually trusting relationship between the doctor and the patient | then patients will feel more empowered and supported in engaging in conversations with their designated doctor | |
| 4. If doctors have the skills, confidence, and inter-personal relations to have conversations with patients about goals of care in acute illness | then they will be more effective in communicating and providing the opportunity and power for patients to speak openly about goals of care in severe acute illness | |
| 5. If healthcare professionals can identify patients who are most likely to benefit from balanced goals of care conversations in severe acute illness | then they will prioritise speaking to patients deemed to benefit most from goals of care conversations | Which will more likely allow a shared understanding between doctor and patient relating to treatment goals and priorities in severe acute illness |
| 6. If healthcare professionals value the importance and acknowledge the benefits of a patient-centred care approach in severe acute illness | then they will be incentivised and motivated to engage with patients | |
| 7. If there is a better understanding of organisational factors that promote or inhibit goals of care conversations between doctors and patients in severe acute illness | then changes can be made and systems developed at the organisational level that facilitate doctors to initiate conversations with patients about goals of care | |

**Relevance screening.** Articles were screened for relevance (eligibility) based on their ability to contribute to the evidence of theory building (CMOs) and the study's initial aims and objectives [30]. The inclusion criteria included articles that had any relevance to any of the CMOs. This was purposely kept broad to reduce the risk of paper selection bias. There were no absolute exclusion criteria as this allowed for important themes to be extracted that might be highly relevant to the programme theory, for example, if they were outside the context of the acute hospital setting but thought to be highly applicable to our study aim (e.g. ethics or behavioural science studies). Exclusion criteria included clinical-based studies: (i) that did not involve adult patients (adults defined ≥as 18 years of age), (ii) that took place exclusively outside the hospital setting (e.g. primary care) (iii) where patients were not admitted to hospital due to acute illness (e.g. hospital outpatients, rehabilitation centres) and iv) did not involve a direct conversation with the patient. For any article meeting any relative exclusion criteria, but still considered to be highly relevant to programme theory building an agreement was sought between JG and JK. Relevance screening conformed to two stages: (i) title and abstract alone, (ii) the whole article.

**Rigour screening.** The final screening process involved critical appraisal of the evidence using well-established quality appraisal checklists developed by the Joanna-Briggs Institute (JBI) [36]. JBI appraisal tools were used for analytical cross-sectional studies, case series, cohort studies, qualitative studies, quasi-experimental studies, randomised control trials, systematic reviews, text and opinion articles. For ethics-related studies, Jansen and Ellerton's Ethics critical appraisal worksheet [37] was used and the Mixed Methods Appraisal Tool for mixed methods studies [38]. Each article was assessed against each of the criteria of the appropriate checklist according to the study design, to give a final absolute score and a relative score as a percentage of the total number of scoring criteria assessed against (S2 Appendix). Articles

were given a final quality grade rating depending on relative scoring ranges of less than 60% (low), 60–79% (moderate) and greater than or equal to 80% (high) (Table 3). Expert opinion and ethical argument papers were all rated as low-quality grade rating irrespective of the absolute or relative scores. This is consistent with internationally recognised guidance of hierarchal evidence [39, 40], where expert opinion has the highest risk of bias and therefore thought to be the least trustworthy for this review.

## 4. Extracting data and synthesis of findings

Data analysis and synthesis processes were flexible, iterative, and creative. To maintain transparency, JG and JK kept notes from a series of meetings during which they discussed each article and its contribution to the CMOs. We used abductive reasoning for the non-observable data to create associations and to recontextualise the data, creating new plausible conclusions [30, 41]. Moving between theory and data, we used retroduction to explore, compare and explain observable patterns in data, whilst also looking for other relevant themes not captured by initial programme theories. Abductive reasoning was used to create associations between theories. For both processes, JG and JK discussed potential explanations, new findings, and strategies to refine and revise the CMOs. We retained notes and a schematic as an audit trail of decisions made. The final synthesis represents an interpretive, yet robust collation of the supporting evidence located for each of the CMOs.

## Results

Following de-duplication, relevance and rigour screening, 52 articles were selected for the final realist synthesis. The data screening processes using the Preferred Reporting Items for Systematic Reviews and Meta-analyses (PRISMA) are depicted in Fig 1. Articles were principally from the USA and European countries (Table 3). Article types represented were qualitative (n = 27), randomised controlled trials (n = 2) observational analytical (n = 3), quasi-experimental (n = 2), cohort (n = 1), mixed methods (n = 5), systematic reviews (n = 3), ethical debate (n = 1), guidance documents/non-systematic review and expert opinion pieces (n = 8). Of the three included systematic reviews, only one of the reviews contained two articles featured in this review [42] (Deep et al, 2008 [43, 44]). Table 3 summarises study characteristics which include evaluation of relevance and rigour analysis. To promote transparency, the data presented in Table 4 are direct quotations from the supporting literature [30].

## CMO One: Information provided to patients in a personalised and acceptable way that allows them to fully understand about their illness, the benefits and burdens of treatment options and potential outcomes, will ensure they are more informed to be able to participate in more meaningful goals of care conversations

Information provided to patients and or their relatives about their clinical condition, likely prognosis and benefits and burdens of life-sustaining and alternative treatments were explored in 26 articles [16, 22, 23, 42, 43, 45, 51, 52, 56, 58, 63, 67, 70, 73–79, 81–86, 89]. Collective analysis of these papers yielded six sub-themes that related to (i) patient preconditions to receiving information, (ii) inconsistencies in information provision by a doctor to a patient (iii) personalisation of information, (iv) the role of decision aids, (v) patient understanding of information and (vi) consequences of being poorly or misinformed.

**Table 3. Characteristics of included studies.**

| Author | Country | Article type | Sample Size | Aim of Study | Relevance of study to research question (rated 'low', 'moderate' or 'high') and methodological rigour (rated 'low', 'moderate' or 'high') |
|---|---|---|---|---|---|
| Anderson et al 2011[45] | USA | Cross-sectional observational analysis | Patients (n = 80) Doctors (n = 27) | To determine whether attending hospital doctors' discussions conforms to recommendations by professional associations and bioethicists | High/Mod |
| Ashana et al 2022 [46] | USA | Qualitative | Doctors (n = 49) Nurses (n = 12) Social Worker (n = 10) Chaplain (n = 3) | To explore facilitators and barriers to having ACP conversations in structurally marginalised groups | High/Moderate |
| Bedulli et al 2023 [47] | Switzerland | Qualitative | Doctors (n = 19) | To explore obstacles to patient inclusion in CPR/DNACPR decisions and challenging conversations. To qualitatively explore physician-reported CPR/ DNAR decision-making approaches and CPR/ DNAR conversations with patients | High/Moderate |
| Bristowe et al 2015 [48] | UK | Mixed-methods | Patients: Quantitative (n = 95) Qualitative (n = 19) | To examine the experience of care supported by the AMBER care bundle compared to standard care in the context of clinical uncertainty, deterioration, and limited reversibility. | Low/Moderate |
| Brooks et al 2018 [49] | USA (n = 5) Belgium (n = 1) Iran (n = 1) Multiple (n = 2) | Systematic review | Studies (n = 9) (None included in realist analysis) | (i) To describe whether culturally sensitive communication is used by clinicians (nurses and physicians) when communicating with patients and families at the end-of-life in the intensive care unit and (ii) To evaluate the impact of culturally sensitive communication at the end-of-life | Mod/Moderate |
| Carrard 2018[50] | Switzerland | Mixed methods | Doctors (n = 61) Patients (n = 244) | To explore the concept of physician behavioural adaptability and how this may be linked to positive consultation outcomes with patients | High/High |
| Casteneda-Guarderas et al 2016[51] | USA | Non-systematic review and expert opinion | N/A | Exploring the issue relating to shared decision-making with vulnerable populations in the emergency department and making a case for the research agenda | Moderate/Low |
| Charles et al 2006 [52] | Canada | Expert opinion based on experience and non-systematic literature review | N/A | To describe the influence of culture on decision-making in the patient-physician encounter and describe how culture impacts the effectiveness of decision aids | High/Low |
| Deep, et al 2008 [43] | USA | Qualitative | Interviews (n = 56) Doctor-patient/ family dyad (n = 28) | To explore how discussions about life-sustaining treatment occur and examine the factors that influence doctors' communicative practices | High/Moderate |

(*Continued*)

**Table 3.** (Continued)

| Author | Country | Article type | Sample Size | Aim of Study | Relevance of study to research question (rated 'low', 'moderate' or 'high') and methodological rigour (rated 'low', 'moderate' or 'high') |
|---|---|---|---|---|---|
| Deep, et al 2008 [44] | USA | Qualitative | 56 interviews with 28 doctors/surrogate dyads | To explore how seriously ill hospitalised patients, their family members and physicians interpret the discussion of the patient's preferences for CPR | High/Moderate |
| Deptola et al 2018 [53] | USA | Quasi-experimental | Patients (n = 283) | To explore whether an intervention to prompt goals of care conversations for those towards end-of-life reduces delays ICU admissions and improves goals of care conversations | Moderate/Low |
| Dubov 2017[54] | USA | Ethical argument | N/A | To provide an ethical argument as to the appropriateness of persuasive communication in critical care and the context of shared decision-making | High/Low |
| Dzeng et al 2015 [55] | USA and UK | Qualitative | Doctors (n = 58) | To explore how physicians' approaches to DNACPR decision-making at the end of life are shaped by institutional cultures and policies surrounding patient autonomy | High/High |
| Eli et al 2021[22] | UK | Qualitative | Doctors (n = 34) | To understand why, when, and how ReSPECT conversations unfold in practice | Low/High |
| Griffiths et al 2020 [56] | UK | Qualitative: ethnography with semi-structured interviews | Doctors (n = 73) | To explore the factors that underpin decisions to admit (or not) to the ICU | Low/ Moderate |
| Haliko et al 2018 [57] | USA | Qualitative | Doctors (n = 73) | To explore thought processes when encountering a simulated critically and terminally ill elder and to compare those models based on whether their treatment plan was patient preference-concordant or preference-discordant | Low/ Moderate |
| Harris et al 2021 [58] | Australia | Qualitative | Patients (n = 10) Family (n = 2) Doctors (n = 4) | To explore patient and family experience of goals of care discussions in hospital within 72 hours of hospital admission | High/High |
| Hart et al 2021[59] | USA | Mixed-methods | Physicians Quantitative (n = 93) Qualitative (n = 15) | To assess doctors' abilities to predict how common choice frames and biases influence people's choices | High/High |
| Hayes et al 2010 [60] | Australia | Qualitative | Total participant size (n = 33) Doctors (junior) (n = 11) Doctors (senior) (n = 11) Nurses (n = 11) | To explore the role of trust in decision-making about cardiopulmonary resuscitation | High/High |

(*Continued*)

**Table 3.** (Continued)

| Author | Country | Article type | Sample Size | Aim of Study | Relevance of study to research question (rated 'low', 'moderate' or 'high') and methodological rigour (rated 'low', 'moderate' or 'high') |
|---|---|---|---|---|---|
| Hutchison et al 2016[61] | USA | Qualitative | Surrogates (n = 30) | To identify dimensions of trust and clinician behaviours conductive to trust formation in relatives of intensive care patients following ICU discharge | Low/High |
| Kon et al 2016[62] | USA | Expert opinion | N/A | To provide a consensus statement to define shared decision-making, recommend when it should be used, identify the range of ethically acceptable decision-making models and present important communication skills | Moderate/Low |
| Kryworuchko et al 2016[63] | Canada | Qualitative | Total participant size (n = 30) Doctors (junior) (n = 9) Doctors (senior) (n = 9) Nurses (n-12) | To identify factors influencing communication and decision-making, and to learn how doctors and nurses view their roles in deciding about the use of life-sustaining technology for seriously ill hospitalised patients and their families | High/ Moderate |
| Lagrotteria et al 2021[64] | Canada | Qualitative | Total participants (n = 23) Doctors (n = 19) Nurse practitioners (n = 3) Social worker (n = 1) | To explore doctors' experiences and perceptions of the Serious Illness Conversations Programme (SICP), a multifaceted capacity-building intervention to improve communication with patients who are seriously ill | Moderate/High |
| Loewenstein 2005 [65] | USA | Expert opinion | N/A | To explore how hot-cold empathy gaps affect preferences and behaviour | High/Low |
| Lee et al 2022[66] | USA | Randomised Control Trial (pilot study) | Patients (n = 150) | To evaluate the efficacy, feasibility, and acceptability of a patient-facing and doctor-facing communication-priming intervention to promote goals-of-care communication for patients hospitalised with serious illness | Low/ Moderate |
| Levinson et al 2019[67] | Australia | Qualitative | Doctors (n = 18) | This study aims to describe the opinions of doctors in emergency departments and how they undertake goals of care conversations with acutely unwell emergency patients and/or their families. | Low/ Moderate |
| Lindberg et al 2015[68] | Sweden | Qualitative (n = 11) | Patients (n = 11) | To describe and elucidate patient experiences of autonomy in an intensive care context from a caring perspective | Low/ Moderate |
| Lu et al 2015[69] | USA | Qualitative | Doctors (n = 114) | The study objective was to describe the language used by doctors when discussing treatment options | High/ Moderate |

*(Continued)*

**Table 3.** (Continued)

| Author | Country | Article type | Sample Size | Aim of Study | Relevance of study to research question (rated 'low', 'moderate' or 'high') and methodological rigour (rated 'low', 'moderate' or 'high') |
|---|---|---|---|---|---|
| Mentzelopoulos et al 2021[19] | Europe (multiple European nations) | Guidelines based on committee of experts in the field | Experts (n = 12) | To guide the ethical routine practice of resuscitation and end-of-life care in children and adults | Moderate /Low |
| Periyakoil et al 2015[70] | USA | Mixed methods | Clinicians Qualitative (n = 29) Quantitative (n = 1040) | To identify barriers faced by doctors (if any) in conducting effective end-of-life conversations with ethnically diverse patients | High/Low |
| Pham et al 2008 [71] | USA | Qualitative | Interpreter (n = 10) | To what extent due alterations occur during language interpretation involving end-of-life discussions | Moderate/High |
| Pollack et al 2019 [72] | USA | Randomised control trial (pilot) | Doctors (n = 15) Patients (n = 428) | To investigate whether electronic alerts combined with communication skills coaching improved the uptake and quality of goals of care | Low/Low |
| Rasmussen et al 2018[73] | Canada | Qualitative | Patients (n = 4) Family (n = 4) | To investigate the experiences of patients with chronic illness, or their families, in any type of discussions related to advance care planning in the hospital setting before ICU admission | Moderate/Low |
| Ros et al 2021[74] | Netherlands | Observational prospective cohort study | Patients (n = 3410) Doctors (n = 6) | To explore the usefulness of "the surprise question (SQ)" to determine ICU outcome. | Low/High |
| Schonfeld et al 2012[23] | USA | Qualitative | Clinicians (n = 32) | To explore the challenges that are associated with end-of-life conversations in elderly patients with multiple morbidities. | Moderate/High |
| Shah et al 2016 [75] | Canada | Qualitative | Doctors (junior) (n = 15) | To observe how residents (junior doctors) are engaging in goals of care discussions with patients and identify thematic patterns that inhibited (barriers) and promoted discussion (facilitators) about goals of care | Moderate/ Moderate |
| Sharma et al 2014 [76] | USA | Mixed methods study | Doctors (n = 56) | To determine whether a multi-faceted teaching intervention improved the quality of code status discussions | Moderate/Low |
| Sterie et al 2021 [77] | Switzerland | Qualitative | Doctor-patient dyads (n = 43) | To explore the circumstances in which doctors explain CPR as well as their content and the way these explanations are delivered to patients | Moderate/High |
| Strachan et al 2018 [78] | Canada | Qualitative | Doctors (n = 18) Nurses (n = 12) | To critically examine nurses' and doctors' perceptions of the nurse's role in goals of care communication with seriously ill patients and their families | Moderate/High |

(*Continued*)

**Table 3.** (Continued)

| Author | Country | Article type | Sample Size | Aim of Study | Relevance of study to research question (rated 'low', 'moderate' or 'high') and methodological rigour (rated 'low', 'moderate' or 'high') |
|---|---|---|---|---|---|
| Sullivan et al 1996 [79] | Canada | Qualitative | Doctors (n = 15) | To understand patient-doctor information exchange, including the timing of discussion, its initiation and content for patients with chronic obstructive pulmonary disease (COPD) | Moderate/ Moderate |
| Syed et al 2017[80] | Pakistan | Questionnaire based cross-sectional study | Doctors (n = 77) | To explore doctor-reported barriers to code status discussions | Moderate/ Moderate |
| Taylor LJ et al 2018[81] | USA | Qualitative | Clinician-patient/ surrogate dyads (n = 17) Recordings analysed (n = 31) | to explore the patterns of communication extrinsic to a decision aid that may impede goal-concordant care for patients with acute surgical illness. | Low/High |
| Thomas et al 2021 [82] | USA | Expert opinion | N/A | To review the ways in which the practice of shared decision-making can be expanded in a practical sense | High/Low |
| Tulsky et al 2017 [83] | USA + Australia | Expert consensus opinion | Experts (n = 10) | To review evidence base surrounding communication between healthcare professionals and patients living with serious illness and provide a research agenda | Moderate/Low |
| Uy J. et al 2013 [84] | USA | Qualitative: analysis of transcribed simulation encounters | Doctors (n = 98) | To describe variation in hospital-based doctors' communication behaviours and decision-making roles for ICU admission and intubation decisions for an acutely unstable critically and terminally ill patient. | High/Moderate |
| Vaderhaeghen al 2019[42] | USA (16/23) Germany (n = 2), Canada, New Zealand, Switzerland, South Korea, Israel (n = 1) | Systematic review | Studies included (n = 23) | To explore facilitators and barriers for hospitalists (doctors) to have advance care planning conversations | High/High |
| Vanderhaeghen et al 2019[85] | Belgium | Qualitative | Patients/families (n = 29) | To gain an in-depth understanding of patients' and their families' experiences of advance care planning in the hospital setting and their views of facilitators and barriers | High/Moderate |
| Visser et al 2014 [86] | USA (n = 14); Canada (n = 5); Europe combined countries (n = 4); Germany (n = 3); UK (2); Australia, Poland, China, Greece, Austria, Ireland, Hungary, West Indies (n = 1 each) | Systematic review with qualitative synthesis | Papers included (n = 36) | To describe doctor-related barriers to adequate communication within the team and with patients and families, as well as barriers to patient- and family-centred decision-making, towards the end of life in the ICU | Moderate/High |
| Vitale et al 2020 [87] | Italy | Expert opinion | N/A | To describe how fake news may undermine the doctor-patient/ family relationship and negatively impact on communication and decision making | Moderate/Low |

(*Continued*)

**Table 3.** (Continued)

| Author | Country | Article type | Sample Size | Aim of Study | Relevance of study to research question (rated 'low', 'moderate' or 'high') and methodological rigour (rated 'low', 'moderate' or 'high') |
|---|---|---|---|---|---|
| Weigl et al 2009 [88] | Germany | Observational analytical | Doctors (n = 35) | To explore the proportion of time spent with direct patient contact vs time spent multitasking during non-patient contact | Low/Low |
| Wubben et al 2021 [16] | Netherlands | Qualitative | Total participants (n = 29) Doctors (n = 7) Nurses (n = 5) Patients +/- family (n = 9) | To identify views, experiences, and needs for shared decision-making SDM in the ICU according to ICU doctors, ICU nurses and former ICU patients and their close family members. | Low/High |
| You et al 2019[89] | Canada | Quasi-experimental: before and after intervention study | Total participants (n = 71) Patients (n = 43) Family (n = 28) | To evaluate the acceptability and potential effectiveness of a video that provides information about CPR aimed at facilitating shared decision-making about CPR. | Low/ Moderate |

Abbreviations

CPR–cardiopulmonary resuscitation

DNACPR–Do not attempt cardiopulmonary resuscitation

ICU–Intensive Care Unit

SDM–Shared decision making

USA–United States of America

### i) Preconditions to receiving information

The patient's or their family's perception that the content discussed is considered to apply to them was an important prerequisite when engaging in any goals of care conversation [22, 58]. This was particularly relevant for patients at the end-of-life. Elderly patients with frailty and multiple comorbidities were less likely to relate to conversations relating to end-of-life care due to a lack of awareness that they are more likely to be in the last phase of life compared to those with a clearer end-of-life trajectory (e.g. people with cancer) [23].

### ii) Inconsistencies in the content of information provided to patients

Inconsistencies in the content of information provided to patients were common and related to the variability of doctors' explanation of a patient's clinical condition, prognosis and the risks versus benefits of different treatment options [45, 84]. Conversations relating to do-not-attempt cardiopulmonary resuscitation (DNACPR) were more common than other life-sustaining interventions. Prognostic uncertainty was perceived by doctors to be difficult to manage and communicate [16, 86]. The words "death" and "dying" were infrequently mentioned [45, 79] and implied rather than expressed [79]. Moreover, palliative care-related options were not as readily discussed [45] and in some cases only discussed after patients had expressed wishes to forgo life-sustaining care [45]. Information provision was incomplete when doctors perceived the risk of information 'overload' or the risk of confusing the patient, particularly when the doctor considered that the topic of conversation was not immediately relevant to the patient [22]. Information provided was also considered to be inadequate if the doctor sensed

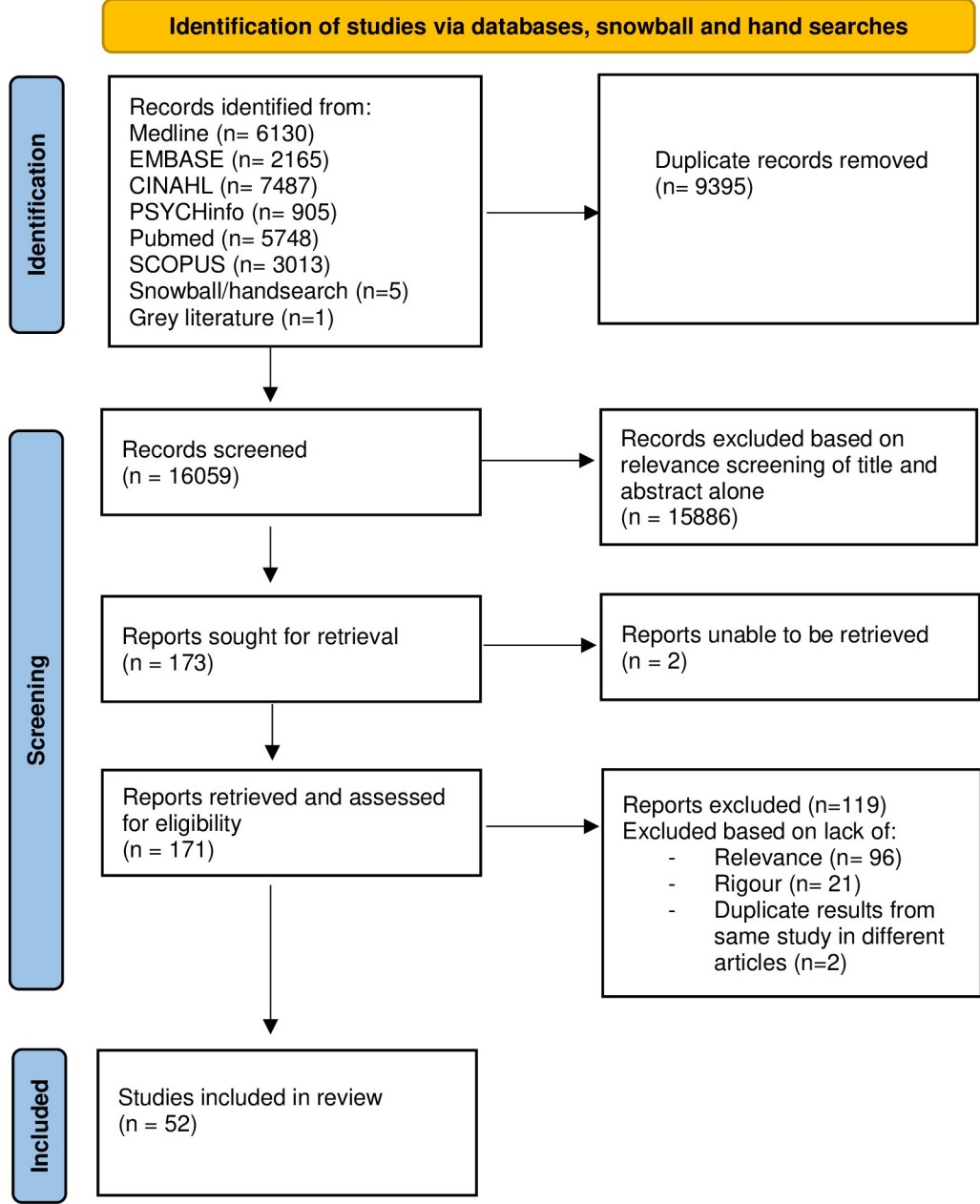

**Fig 1. Preferred reporting items for systematic reviews and meta-analysis (PRISMA).**

rising tension or conflict with the patient and or their family due to disagreements or divergent views between them and the patient or family [22].

## iii) Information not being provided in a personalised manner

Information provided by doctors commonly focussed more on medical or procedural interventions and less on broader life values [42, 43, 45, 47, 56, 58, 75, 77, 82]. Factors associated with this included the doctor's perception that the patient was not at the end-of-life [67, 76], when a medical intervention or surgical procedure was proposed [81], the perception by the

**Table 4. Results summary with illustrative quotes.**

| CMO | Themes and Subthemes (Themes in bold) | Studies relating to theme– 1st Author (Relevant extracts in Italics) |
|---|---|---|
| 1: Information provision | **Preconditions to receiving information (willingness to engage in goals of care discussions)** | Eli [22], Harris [58], Schonfeld [23] |
| | Less willing to engage in conversation if they feel not applicable to them | "… I don't think I was bad enough…health wise…to warrant that sort of situation…I thought no, but I went along with it because you do that when you are in hospital. You go along with things" (Harris [58]); "…I don't like the idea of associating people coming to the hospital with having a discussion about their mortality…I think it's not part of life…death is the end of life…life is about living it…it really sort of plants unpleasant seeds, unnecessary seeds of the end of life." (Harris [58]) |
| | Less perceived applicability in those with no end-of-life diagnosis (e.g. multimorbidity compared with cancer) | "But people will seem to think there's a cure for everything but everyone. But people can somehow accept that 'I have cancer, I'm going to die.' But everything else it seems like 'Well, why don't you fix that problem' like 'There's medicines, there's stuff to do.' And they're not nearly as accepting of any other diagnosis." (Schonfeld [23]) |
| | **Inconsistencies to the nature of information provided to patients** | Anderson [45], Eli [22], Ros [74], Sterie [77], Sullivan [79], Tulsky [83], Uy [84], Visser [86], Wubben [16] |
| | Inconsistencies relating to communication of uncertainty relating to prognosis | "Again, the problem is always that uncertainty. The other day we lost a patient after 6 weeks of treatment. (…)On the one hand you can say that we all saw that coming. On the other hand—well, you only know for certain if you've tried it" (Wubben [16]) |
| | Death/Dying infrequently mentioned and often implied rather than expressed | But we don't stress too much the word "death" or "dying." "I think it becomes evident as we speak" (Sullivan [79]) |
| | **Information not being provided in a personalised manner** | Anderson [45], Bedulli [47], Deep [44], Castaneda-Guarderas [51], Griffiths [56], Harris [58], Krywuruchko [63], Levinson [67], Shah [75], Sharma [76], Sterie [77], Taylor [81], Thomas [82], Tulsky [83] Vanderhaeghen [42], Wubben[16], You [89] |
| | Procedural intervention discussion favoured over broader life goals | "medications that we can give to increase your heart rate, decrease your heart rate, or get it out of an irregular rhythm," "electrical paddles," and "pounding on their chest." (Deep [44]); "…she understood what resuscitation meant…but she didn't clearly understand what intubation and non-invasive ventilation meant… What do those mean in context?" (Harris [58]): |
| | Recognition that there is more of a need for a personalised approach | "Physician: So, let's say that when we have a cardiac attack which… which… in which the heart stops. It's quite serious as complication, so what we have to do is use perhaps some electric current to shock the heart, to put a tube to support the lung. Sometimes one recovers well, other times we can have side effects, or not recover or have side effects. This is something that is unpredictable. Patient: Yes, we can't know. Physician: We can't say what a person… we know that the longer we try to resuscitate… Patient: Yes, yes. Physician: The more possible it is that complications arise, but again we can't know… Patient: No, we can't know, of course. |
| | Doctors currently have preference in discussing medical interventions over broader life goals/QoL. When QoL was brought up by patients, doctors didn't always ask patients to elaborate further | Physician: So, if it happened and if there was an attack, do we try to resuscitate? Patient: Yes try, yes try". (Sterie [77]) |
| | Doctors did not always recognise that patients may view quality of life differently to them | "We know everything medically about them, but we don't know their story and we don't know what informs the decisions they've made to this point" (Krywuruchko [63]) |
| | **The role of decision aids** | "We are not so attentive to the needs of the patient because we, too often, focus on treating the disease and not on treating the person." (Bedulli [47]) |
| | Allows more consistent information to be provided but not culturally sensitive | "Look, in the end we all prefer talking about the fluid balance and CRP levels. That's the truth. So [talking about quality of life] is 'soft' drivel to many people" (Wubben [16]); "PATIENT: My basic position has been that I do not want my life extended…[there is a] serious question as to whether I would be able to return to a relatively normal or natural life. I don't want any artificial means taken to extend my life, if it's not going to extend the quality of my life. PHYSICIAN: I see. I ask because some people absolutely don't want anything done. And other people say, "Do everything." And there's a lot of gray in between. So, I want to make sure we respect your wishes. But I think I have a sense of where you're coming from." (Anderson [45]) |
| | **Understanding of information** | "What I used to see, and still see a bit—is that we physicians have our own opinions about what constitutes a good quality of life—in other words, what a good outcome looks like. And we do not look at the patient well enough" (Wubben [16]) |
| | Acute illness is associated with cognitive impairment and limits patient understanding and meaningful engagement | Castaneda-Guarderas [51], Charles [52] |
| | Poor understanding or misunderstanding is common and occurs when the information provided to patients/families exceeds their information capacity (i.e. their ability to retain and understand such information) | Anderson [45], Castaneda-Guarderas [51], Griffiths [56], Harris [58], Periykoil [70], Rasmussen [73], Shah [75], Strachan [78], Sterie [77], Thomas [82], Tulsky [83], Vanderhaeghen [42], Vanderhaeghen [85] |
| | Prior experience (either as a patient or relative) improved level of understanding | "In the emergency department: The patient looks grey, having difficulty talking, out of breath, panic in their eyes, takes the oxygen mask off repeatedly, seems to be in agony…the junior ICU registrar asks the patient rapid questions about functional status, to which he says yes giving the impression he is in good health…the ICU registrar asks the wife and daughter outside the resuscitation room about the patient's functional status and they give an a more pessimistic picture compared to the patient's (field notes); In the emergency department: Patient seems uncomfortable, twisting in the bed. The CPAP machine is loud. Several staff around the bed doing procedures, such as blood sampling. The senior ICU registrar asks in a loud voice: "Have you thought about what would happen if you get worse?" [No audible answer] "Do you want dialysis?" The patient says "yes"… "Would you want to be put to sleep on a breathing machine?" Patient rotates her hands as it is trying to say she doesn't know." (Griffiths [56]) |
| | **Consequences of being poorly or misinformed** | "…because the family a lot of times is quite overwhelmed and then has a lot of follow-up questions afterwards" [nurse interview] (Strachan [78]); "Certain medical terms may be difficult to explain in a way the patient can understand." "They may not be used to the health system they find themselves in and it may be overlooked that they lack what we would consider common knowledge" "Incomplete understanding of what resources/ therapies that can be versus should be provided for a patient." (Periyakoil [70]) |
| | Increased likelihood of fluctuating patient preferences, conflict between doctor and patient/family and risk of patient discordant treatment | "I don't want to be on a breathing machine. I just went through this with my mom, and I think this is why I'm having a hard decision. Because she passed away in April." (Shah [75], Sterie [77]) |
| | | Deep [44], Shah [75], Sterie [77] |

*(Continued)*

**Table 4.** (Continued)

| CMO | Themes and Subthemes (Themes in bold) | Studies relating to theme– 1st Author (Relevant extracts in Italics) |
|---|---|---|
| CMO 2: Bias and external threats to shared decision making | **A) Influence by doctors** | Brooks [49], Dubov [54], Castaneda-Guarderas [51], Haliko [57], Thomas [82], Visser [86] |
| | **Intuitive decision-making is common when making decisions about life-sustaining treatments in acute settings** | Dubov [54], Dzeng [55], Harris [58], Hart [59], Lu [69], Sullivan [79], Vanderhaeghen [42] |
| | Intuitive decision-making is prone to error and bias | "For patients that I think should be DNR, I go into graphic detail pretty aggressively that we can do chest compressions which can break ribs and puncture lungs, which can be very painful, and we can put them indefinitely on a machine that could prolong their life without improving their quality of life. Then I usually say, 'but of course it is your decision and it should be what you think they would want" (Dzeng [55]) |
| | **Conversation framing bias and persuasion** | |
| | Doctors commonly frame conversations with the intention of seeking to gain patient/family agreement with the management plan. This limits/invalidates patient autonomy | "I won't get into representing how sick you are, but instead say, 'would you like us to pound on your chest and break your ribs.' They are infusing it with such aggressive language that there is a right answer…and it's potentially not an accurate way to frame it…It is so laden with bias that you're taking away the patient's autonomy but still have the illusion of giving full autonomy to them" (Dzeng [55]) |
| | **Fears of repercussions from goals of care conversations** [(increased patient distress, fears of difficult reaction leading to conflict, medico-legal repercussions] | "Depending on how you're presenting [choice options], you're going to influence patients' decisions. But that's kind of your job in the role of an expert consultant, is to influence or recommend. I think most people when they are influencing peoples' decisions are doing it from the place of trying to do what they think is best for the patient. I think that is always ethical…" (Hart [59]) |
| | Doctors are more hesitant about withholding life-sustaining treatment when there is fear of repercussions | "…probably the big influence was the way I sold it to him…sometimes I'll push my agenda more obviously if I disagree with them to try to get them to my same page… it's [medical intervention] not the best thing for people sometimes, but they think it is." (Harris [58]) |
| | **B) Influence by families** | "You might colour it one way to the other, we always do. There are some people who you just know are bad intubation candidates and for those people I'll paint a bleaker picture."(Sullivan [79]) |
| | **The role of family members in goals of care discussions is variable and may be reflective of underlying cultural values.** | Syed [80], Vanderhaeghen [85] |
| | Most patients value the importance of their families being involved | Charles [52], Harris [58] |
| | | "I think it's a joint decision…I would like to do it in conjunction with my kids." [Patient] (Harris [58]) |
| | Doctors sometimes seek to discuss goals of care with patient away from family without exploring whether or not the patient wishes their family to be involved | "In that situation I think I'd rather have my family, I really do. I know it might be upsetting for them but I still would like them to be involved." (Patient] (Harris [58]) |
| | **Discordant views between patient and family members and family dynamics** | "She was competent enough to make the decision…I didn't feel that a family member needed to be there or needed to be contacted because [the patient] can make her own decisions." (clinician](Harris [58]) |
| | Family sometimes more assertive than patients and tend to focus more on survival without full consideration of the trade-offs involved in life sustaining treatments | Brooks [49], Deep [43], Griffiths [56], Rasmussen [73], Shah [75], Syed [80], Vanderhaeghen [85] Patient: "Just let me go…I just don't want any life support." Wife: "Naturally if your heart quits beating, they asked if you want revived. I would want him revived, but not put on life support." Interviewer (to patient): "And you would not want to be revived?" Patient: "I don't think so." At various times throughout the interview, the patient's spouse responded on behalf of the patient or interjected her preferences about his care (Deep [43]) |
| | Complex family dynamics may relate to a family member being more domineering and assertive with their views which threats a breakdown in trust and communication between patient, family and doctor | Patient: "I have said…if I was really ill, a burden, haven't got anything going for me, I would rather not (go to ICU)…but my son says don't talk so silly…he doesn't like talking about things like that" (Griffiths [56]). Patient: 'I didn't talk about my wish for euthanasia with my daughters…I know they will be very sad.'; Husband: 'I know my wife wants to stop when she has to receive haemodialysis. But I think she should give it a try. I still want her around, you know!' (Vanderhaeghen [85]) |
| | **Impact of emotion and low health and information literacy** | "I really struggled at the start of all this with some family input, and it was like "You can't make that decision, you don't know what he would want!" You know, so I found that really hard. And it was like: "I live with him, I know what he wants." And you know, I get what he wants. I know he wants to live". (Family] (Rasmussen [73]) |
| | Hot emotional state (patient or family] makes conversation is stressful | Dubov [54], Kyworuchko [63], Loewenstein [65], Rasmussen [73], Schonfeld [23], Syed [80], Thomas [82], Tulsky [83], Vanderhaeghen [85] |
| | Hot emotional state limits the ability to understand and retain information. This is exacerbated in those with low health and information literacy | "We were blindsided by the whole shock of being here, and then the shock of that conversation. That seemed to happen so quickly"; "Nobody likes to talk about dying, but that's…less stressful talking about something before it happens, opposed to being on your death bed being asked to sign something." (Patient] (Rasmussen [73]) |
| | **C) Other influencers on information provided** | Hayes [60], Tulsky [83], Vitale [87] |
| | **The media, false information, and inconsistencies in information from previous discussions with doctors** | "Television shows…they see all this magical stuff done, and when it comes to their loved one…they think there should be no reason why this magical stuff shouldn't be able to be done. I think there are often unreasonable expectations" (Senior doctor]. "No longer does it take 6 years basic training as a doctor and specialist training for years after to make that decision, to be armed with the information. Whereas they can look it up in five minutes on the internet, the condition, and become instant experts…and I've had the conversation…and it's almost like they're waiting, they're trying to trip you up" (Nurse] (Hayes[60]) |
| | **Language translator** | Periyakoil [70], Pham [71], Syed [80] |
| | Language translators can introduce biases, or alter conversations | "Clinician: I don't know. Um, this is a very rapidly progressing cancer. Interpreter (translating] He doesn't know because it starts gradually; Clinician: The problem with this option is that he may have to stay on this machine for the rest of his life. Interpreter (translating] But the problem with this option is that he will have to stay on this machine for the rest of his life" (Pham [71]) |

*(Continued)*

**Table 4.** (Continued)

| CMO | Themes and Subthemes (Themes in bold) | Studies relating to theme– 1ˢᵗ Author (Relevant extracts in Italics) |
|---|---|---|
| **CMO 3 and CMO4: Skills and confidence of doctors for engaging in goals of care conversations and trusting relationships with patients and families** | **A) Factors associated with a lower level of trust** Preconceived views and stereotyping• Low cultural literacy amongst HCPs• Power differential between patient/family and doctor and avoidance of potential conflict Other factors associated with low level of trust (negative prior consultations, lack of openness and honesty, language barriers, fear of abandonment)**B) Factors associated with a higher level of trust** Appropriately skilled doctors• Perceived credibility of doctors• Being personable• Empowering patients/families• Providing information 'at the right level" for the patient/family and avoiding medical jargon• Continuity of care**C) Consequences of mistrust**• Breakdown in relationships between doctor and patient/family leading to conflict, mistrust, and more invasive care**D) Lack of Confidence of Clinicians in engaging in goals of care conversations and making treatment recommendations** • Low confidence exacerbated by potential conflict anticipated as a result of unrealistic patient/family expectations • Difficulty managing and communicating clinical uncertainty • Poor understanding by doctors what the most appropriate type of doctor-patient relationship for individual patient • Difficulty with integrating patient's personal values into making a treatment recommendation **E) The role of further training in goals of care communication and proposed future directions in training**• Welcomed by doctors• Most effective training techniques are unknown• No existing validated outcome measures relating to the effectiveness of training intervention**F) Communication guides and complex interventions**• Have the potential to harbour better relationships• May result in more scripted conversations | Castaneda-Guarderas [51], Hayes [60], Vanderhaeghen [85], Vitale [87] Ashana [46], Brooks [49], Castaneda-Guarderas [51], Charles [52], Hayes [60], Periyakoil [70], Syed [80], Thomas [82] *"Doctors not understanding the cultural values surrounding end-of-life care for a patient with a different ethnic/religious background."; "Unfamiliar with social norms for showing sympathy, hug? cry?"; "Cultural norms that differ from my own causing me to inadvertently offend the patient or his/ her family."; "Not understanding which topics might be taboo."; "Not knowing how to discuss goals in a way that makes sense to someone with different views about death based on different beliefs about spirituality and afterlife." (Periyakoil [70])* *"In the Middle East you do things basically until even after death. There's no way you can stop and say—I will not resuscitate or I will not do this or there's nothing more.... People coming to western countries from the Middle East would be quite not accepting of the idea of stopping treatment, their idea is to have treatment... In the Middle East you would never discuss with the patient resuscitation or decision-making (Senior doctor)" (Hayes[60])* Castaneda-Guarderas [51], Kryworuchko [63], Lindberg [68], Schonfeld [23], Syed [80], Visser [86], Wubben [16] *"sometimes it's about us, we're not comfortable making that decision either" (resident physician] and 'And I can say this with certainty, that there are people, and I've seen it with colleagues as well as students, who are afraid of this: who are afraid of talking about anything related to end of life with people.' (Staff physician] (Kryworuchko[63])* *"...That you're scared of promising something you can't fulfil. It's weird to then not ask the question, but that is a way of doing things. Or fearing totally irrational wishes from people." (Wubben[16])* Bedulli [47], Harris [58], Kryworuchko [63], Levinson [67], Pham [71], Vanderhaeghen [85], Visser [86] *'I don't know if the physician is completely open. I feel my body is getting worse and worse. But she brings me good news... I don't know what to believe.' (Vanderhaeghen[85])* *"They have a misconception. They think if you have one of those orders filled in then the doctors don't try as hard to treat them; they won't give them all the other medical treatment that they need (Junior doctor)." (Hayes[60])* *'One of the daughters was angry. She was saying we were abandoning her mother. That we weren't allowed to do that. That we had to keep it up until the end.' (Nurse] (Kryworuchko [63]); "They may think we just let them die like that, but that is simply not what comes next. (FG1]"; "It is frustrating that they may think we want to abandon them" (Bedulli [47])* Carrard [50], Harris [58], Hayes [60], Hutchison [61], Kryworuchko [63], Lindberg [68], Mentzelopoulos [19], Rasmussen [73], Schonfeld [23], Thomas [82], Vanderhaeghen [85], Wubben [16] *"Because somebody has explained to you, you know, that your heart has changed shape, I didn't know that because it's not my area of expertise.... And you have to trust somebody who's got more knowledge than you to make those decisions on your behalf." (Patient] (Harris [58]).* *"Well I think it's a huge thing because they're well... especially if you're telling the patient and their family what you think is the right or wrong thing. I think that it takes a big weight off people's chests and there's a huge element of trust...they're trusting that you know enough about their illness to know what the likely prognosis and what the likely survival is (Senior doctor)." (Hayes[60])* *"And he gave me that glimpse of his personal life, which, I thought, 'Ahh! You do understand. You're not just in a white coat and not a human being. But you get it'... They gave you that little insight of their personal life... I appreciate that." (Hutchison [61])* *'It's okay to do this and I know you're going to feel guilty, but it's okay to make this decision.'' I think that it's okay to forgo treatment, to forgo burdensome treatment. Sometimes they allows them to even discuss it" (Schonfeld [23]);* *"And sometimes I think we just need to empower patients or families. It is okay to forgo treatment, to forgo burdensome treatment. Sometimes they don't realize this." (Schonfeld [23])* *"They explain it to me in a way that I can understand. They gain my trust because they're not leaving pieces behind. I'm understanding what they're telling me. I think that will help you gain trust in a person." (Hutchison [61])* *"... and he made complete sense to me... he talked in enough layman's terms that we could understand... the issues that he was describing" (Rasmussen [73])* Hayes [60], Kryworuchko [63], Schonfeld [23], Strachan [78] *"The aim there is really to try and build rapport... so they end up trusting... particularly if they see us continuing the treatment..." (Hayes[60])* *"We spend a lot more time. I might spend a week with a person, eight hours a day for four or five times in a week and so I think that amount of time and that kind of exposure and relationship sort of leads to more conversations sometimes." (Strachan [78])* Hayes [60] Anderson [45], Bedulli [47], Brooks [49], Haliko [57], Kryworuchko [63], Mentzelopoulos [19], Pollak [72], Sharma [76], Sullivan [79], Syed [80], Taylor [81], Uy [84], Vanderhaeghen [85], Vanderhaeghen [42], Visser [86], Wubben [16] *"It is always difficult to approach. ...It's not always difficult, well, it is not always difficult, but it's a bit uncomfortable approaching that particularly in the presence of the patient, hence the hesitation." (Haliko [57])* *"sometimes it's about us, we're not comfortable making that decision either" (resident physician] and 'And I can say this with certainty, that there are people, and I've seen it with colleagues as well as students, who are afraid of this: who are afraid of talking about anything related to end of life with people.' (Staff physician] (Kryworuchko [63]):* *"...That you're scared of promising something you can't fulfil. It's weird to then not ask the question, but that is a way of doing things. Or fearing totally irrational wishes from people." (Wubben [16])* DOCTOR: "I'm sure it's hard.... I just wanted to clarify a couple of things. One of them is in case of an emergency in the hospital have you talked to a doctor or even a family member about what you would want us to do in case of an emergency? Patient: No. DOCTOR: No, okay. So things like putting a tube down your throat to help you breathe, or antibiotics, things that would help prolong your life. Would you want us to do that while you were here?" (Sharma [76]) Kryworuchko [63], Levinson [67], Mentzelopoulos [19], Sharma [76], Tulsky [83] Lagrotteria[64] *"I think what I found most useful was a separation of time and space; [it] created a moment to build a relationship in a way that we don't always in acute care because we don't either have the time or the dynamic is different."; Getting to know what's important to the patient: "[The guide] creates a completely different environment because you're asking questions about their bigger life values and goals... We don't ask these things. It's really pivotal for me; it really changes my practice." (Lagrotteria[64])* *"I just think just the static nature of it makes it a bit difficult to adapt to an actual, real- life conversation sometimes." (Lagrotteria [64])* |

*(Continued)*

**Table 4.** (Continued)

| CMO | Themes and Subthemes (Themes in bold) | Studies relating to theme– 1st Author (Relevant extracts in Italics) |
|---|---|---|
| CMO 5: Identifying and prioritisation of patients for GoC conversations | **Identifying and prioritisation of patients for goals of care conversations**<br>• Lack of threshold, precipitating or triggering event makes it difficult for doctors to decide when is the most appropriate time to have goals of care conversation<br>• Complex interventions (e.g. SICP) have the potential to identify patients suitable for goals of care conversation at an earlier stage | Bristowe [48], Deptola [53], Eli [22], Lagrotteria [64], Lee [66], Pollak [72], Schonfeld [23], Sullivan [79], Vanderhaeghen [85], Visser [86]<br><br>*"But what, at what hospitalization, at what co-morbidity that pops up, do you then approach them and have this end-of-life conversation?" (Schonfeld [23])*<br><br>*"I think we're doing a better job of identifying these patients earlier, and whether it's attributable to this program, specifically, or this is just supporting a culture shift in that regard, I'm not very sure." (Lagrotteria[64])* |
| CMO 6: Valuing goals of care conversations and motivation and incentivisation to engage | **Doctors value goals of care conversations and motivation and incentivisation to engage**<br>• Doctors who have seen the benefits themselves may be more likely to be motivated to hold goals of care conversations with patients which gives their job more of a sense of purpose<br>• "Path of least resistance" may disincentivise doctors to have goals of care conversations with patients | Lagrotteria [64], Levinson [67], Thomas [82], Vanderhaeghen [42], Wubben [16]<br><br>*"I think we're all so invested in this process because we know it helps and it helps to give better care to our patients"; "I realized how patients are so willing to open up and speak, and it actually does work, some of these specific questions, of areas to explore. It allows me to be more using of it, and it's nice using something that's been validated, researched, so it really adds to the toolbox" (Lagrotteria [64])*<br><br>*"It definitely. . .makes the day feel much more fulfilling to be able to connect with the patient or family on that deeper level as opposed to the more superficial and very busy tasks of the day." (Lagrotteria [64])*<br><br>*"At the same time it's easier for me to intubate, (. . .)to start renal replacement therapy—far easier than not starting treatment. So I think that's an important point. (. . .) Sometimes we use the multidisciplinary discussion to say to each other: are we really still on the right track?(. . .)And then you sometimes get one-liners like: 'You can always stop [treatment], the patient can always say that they don't want it like this [at a later stage].'" (Wubben [16])* |
| CMO7: The impact of hospital-related factors on goals of care discussions | **Organisational factors that may facilitate goals of care conversations**<br>• Organisational culture and policies<br>**Organisational factors that may hinder goals of care conversations**<br>• Time constraints<br>• lack of doctor "ownership"/responsibility for the patient<br>• Lack of appropriate clinical environment to hold goals of care conversations<br>• mistrust in the organisation/medical system<br>• frequent shift changes/lack of continuity of care<br>• level of care following withholding life-sustaining treatment<br>**Suggested organisational changes to facilitate goals of care discussions**<br>• better defined roles for nurses with more valued input<br>• unit champions/administrative support<br>• organisational inclusivity of minority groups | Schonfeld [23], Syed [80]<br><br>Harris [58], Kryworuchko [63], Periyakoil [70], Schonfeld [23], Syed [80], Thomas [82], Visser [86], Weigl [88], Wubben [16]<br><br>*"sometimes it's that we don't have time [to engage in GoC conversations]" (resident physician] (Kryworuchko [63])*<br><br>*"I grant you this often gets overlooked [end of life conversation] in the hectic pace of everything else we do." (Schonfeld [23]): "The limits are mostly put on by time and space. Sometimes you have a really busy day so you don't have time for it. Then you need to cut back a little on those conversations, because there isn't any time." (Wubben [16])*<br><br>*"Some groups feel more marginalized in the community at large and this makes them more distrustful of the medical system as a whole."; "Patients may believe that care is being "withdrawn" "from their loved one because of racism." "Certain cultures lack trust in the medical profession, do not believe physicians have their best interests at heart." "Fears of abandonment or self-interested medical professionals". (Periyakoil [70])*<br><br>Brooks [49], Charles [52], Castaneda-Gaurderas [51], Kryworuchko [63], Lagrotteria [64], Strachan [78], Wubben [16]<br><br>*"There are physicians who will grant that role to nurses, and others who will assume that role. And that's why we never know what our role is. [nurse]" (Strachan [78])*<br><br>*"It's out of my hands whether or not it's taken into consideration or not. You can tell residents all they want but if they have something set in their mind that this is going to happen then that's going to happen. Most of the times we can't change their minds. But you never know.' (Nurse] (Kryworuchko [63])*<br><br>*"ICU Nurse (N): Then after the family conference, you let it sink in, and you start repeating it and repeating it. And you try to use the same words as the physician—because I've noticed families say: I think it's so difficult, one says this and the others says this—but that's because [families] don't understand. N: That is kind of the role we take on: [translating the family's wishes for the physicians]. N: Yes—[nurses] think it's important to be of value in decision moments. Continuing or not, you know. Of course, you need to do so based on medical information, but also based on the holistic view, and I think we should play a larger part in that, because we also know the family really well." (Wubben [16])*<br><br>*They've just been told something potentially devastating. So, you've got to ask how much they actually retained. So that's usually the best place. So, gleaning a bit of an insight into what they understand, what they understand, what this means to them or what they're understanding it means, is probably the biggest step for the nurse to take after they've had that change.' (Nurse] (Kryworuchko [63])*<br><br>*"There's a cueing reminder, there's an administrative burden removed where someone else is scheduling and letting me know when it [the conversation] is, based on my availability, and that really helps. The patients and family are primed on it. It was just any time I wanted to do it, and whenever we planned to do it, it just happened. Everyone was on board, and everything was set up." (Lagrotteria [64])* |

doctor that quality of life was a "softer topic" [16], the perception that medical intervention was easier to communicate than personalised life goals [89]. Language barriers presented additional challenges for doctors in providing personalised information to patients [51]. When quality of life was mentioned by patients, doctors did not always ask patients to elaborate further [45] and often reverted to how they viewed quality of life, which may differ from the quality of life as perceived by the patient.

### iv) The role of decision aids

The purpose of decision aids is to assist patients in understanding their clinical situation and facilitate communication between doctor and patient. Their underlying purpose is to help patients participate in medical decisions that relate to them [51, 52]. Two studies demonstrated that decision aids allowed more consistent information to be delivered to patients. However, this was not specific to the needs of minority ethnic and or vulnerable patient groups [51, 52].

### v) Understanding of information

Patient's understanding of the information provided to them was influenced by their lack of mental capacity due to pre-existing cognitive impairment, for example, dementia [83], the effects of acute illness including, the presence of physical pain or other associated discomfort [56, 58, 73, 82]. For patients who were deemed to have mental capacity, information was often poorly or incompletely understood. This occurred when the information provided by the doctor exceeded the patient's ability or capacity to understand and retain specific items of information. Doctor factors that were associated with information overload included the overuse of medical jargon [42, 45, 73], talking too quickly [58] and providing too much information at a specified point in time [42, 45, 58, 73]. Patient factors that were associated with a lower proficiency for understanding were low health and information literacy [51, 70], high emotional states [78, 82] and language barriers [51]. These all contributed to poor understanding between doctor and patient. Prior patient experience of hospital treatment and doctors checking understanding of any new information provided contributed to improved understanding [47]. Although doctors acknowledged the importance of patient understanding and recognised patients' expectations, wishes and values [47], these were inconsistently explored and patients did not always feel empowered to speak up or own up in the event of a lack of understanding [42]. This resulted in patients being less informed [42].

### vi) Consequences of being poorly or misinformed

Poor or misinformation resulted in poor recollection by the patient of the contents and nature of discussions about the goals of care. This had the potential to contribute to fluctuating patient treatment preferences, increase conflict between patients and their families and doctors and increase the possibility of discordant treatment [43, 75, 77].

### CMO Two: Patients receiving information about treatment options and their potential outcomes that is biased, together with the impact of other external factors, will have an influence on the judgments patients make about their wishes for future care and decrease the likelihood of a shared decision-making approach

### A) Doctors

Intuition often predominates over analytical decision-making by doctors in acute hospital illnesses [54, 57, 82]. Intuition uses experience, feelings and accumulated judgments that

culminate in heuristics (mental shortcuts). These are prone to error and bias which can translate into communication biases [86] or conversation avoidance [49]. Although some doctors acknowledged the importance of remaining neutral, factual and non-influential when providing information [47] this was contrary to what was observed in practice. Conversation-framing bias was reported in eight studies [42, 47, 54, 55, 58, 59, 69, 79] and was more common in situations where higher clinical uncertainty was present [69]. 'Framing' is a cognitive bias that relates to how a patient's decision or thought process may be influenced by the way information is presented to them by their doctor. Evidence demonstrated that doctors sometimes persuade patients to agree with their thoughts about the goals of care [42, 54, 55, 58, 59, 69, 79]. The rationale described was superior clinical knowledge to make better 'best interest' decisions. It was also thought to minimise the influence of highly charged emotions experienced by families concerning acute illness that could potentially contradict the patient's long-term goals [42, 58, 59].

Fears of repercussions were reported as a barrier to having goals of care discussions and were perceived to change the nature of such conversations. These were related to fears of causing undue suffering and distress, fears of a "difficult reaction" from family or patients, leading to conflict and fears of medico-legal repercussions [80]. As a consequence, this resulted in doctors being more hesitant about withholding life-sustaining treatment [85].

## B) Influence by families

Families played a varying role in goals of care conversations. This may be influenced by different personal sociocultural beliefs about illness and the role that they (and their patient relative) were expected to play in such conversations [52]. There were examples of situations where patients viewed their family members as integral to goals of care discussions, sometimes even deferring to them for all discussions and decisions relating to their medical care. However, this was not always identified by their doctors who sometimes made efforts to speak with the patient when family members were not present [58]. Discordant views regarding treatment options also existed between patients and family members [47]. This occurred when patients and their families held different motives and priorities relating to what they viewed as desirable medical treatments and associated outcomes [43, 75, 85], for example, the family prioritising survival whilst underappreciating the trade-offs of physical and mental morbidity (e.g. delirium and psychosis) associated with life-sustaining treatment [85]. Strongly expressed views from a family member were shown to influence a patient's thought processes relating to goals of care [80]. This was particularly relevant when the intra-family conflict was present and had the potential to harm trusting relations between doctors, family and patient [73] culminating in fewer goals of care conversations [49]. Doctors also recalled some of their patients' wishes centred more on what their family wanted rather than them, which in some circumstances contradicted their values [46].

The impact of a family's emotional state was intrinsically linked to the appropriate timing of conversation about goals of care [23, 54, 63, 73, 83, 85]. This was particularly relevant to crises where there was an imminent risk of death [65]. Highly charged emotions were identified as being further intensified by the presence of low health and information literacy among family members [54, 80]. This sometimes resulted in instability of expressed preferences and views based more on feelings and less on facts or balanced opinions [82]. Unrealistic expectations, usually associated with undue optimism relating to prognosis and treatment were also evident [49, 90].

## c) Other influencers on Information provided

The internet and media reporting, for example, the reporting of "miracle cases", may be influential towards patient and family misconceptions about treatments and prognosis before goals

of care conversations [60, 83], potentially leading to mistrust [60, 83]. Prior discussions with doctors that had inconsistent themes were also shown to impact conversations. This was particularly relevant when patients and their families had the desire to "snatch" clinical news from different conversations, hoping to get a custom-made truth that better fitted their emotional desires regardless of their positive or negative value [87]. Language barriers and the use of translators resulted in additional biases, omissions or additions introduced by the interpreter and are thought to have potentially significant consequences on goals of care conversations [71].

## CMO Three and Four: Doctors that have the confidence and interpersonal skills to form more trusting relationships with their patients, will result in patients feeling more supported and empowered to speak more openly about their goals of care in acute illness, leading to a better shared understanding between them and their doctor

The type and strength of the doctor-patient relationship were integral to the development of mutually trusting relationships.

### (A) Skills of doctors (clinical expertise, communication and interpersonal skills) to develop trusting relationships

Appropriately skilled doctors were perceived by patients/families to have credibility, clinical expertise and a high level of competence [58, 60, 61]. Appropriate interpersonal skills were also perceived as important. These included measures such as introducing themselves, addressing patients or surrogates by their name, being personable [47, 72] and warm [61, 85], being approachable [85], listening to and understanding patients' queries and concerns [23, 60, 63], providing information in a clear and jargon-free manner [19, 47, 61, 73, 82], showing honesty [85], compassion [47, 61, 73] and treating patients with respect and dignity [16]. The inclusion of family members in discussions (with the approval of the patient) was also deemed important in certain situations [19]. In situations when a patient lacked capacity, some family members took more of an advocacy role as the patient's spokesperson, sometimes by default. This sometimes resulted in guilt, either expressed or implied by the family, particularly when asked for their input on potentially withdrawing or withholding life-sustaining care for those at the end-of-life. Doctors who acknowledged this and were understanding, supportive and compassionate resulted in more trusting relations and paved the way for families to discuss such issues more openly [23]. Continuity of care associated with its physical presence and frequent communication were also integral to building a relationship, rapport and mutual trust between doctors and patients [19, 23, 42, 60, 63, 72, 79, 85].

However, doctors frequently admitted lacking confidence and skills in discussing goals of care. Fear and anxiety of "taking away hope" and or getting into conflict with patients or families were reported, particularly when the patient and their families were perceived to have unrealistic expectations [16, 42, 79]. Patient or family trust was also perceived by doctors to be threatened when decisions to limit or withhold life-sustaining treatments were suggested, due to fear of abandonment and inferior care [85]. Doctors felt poorly equipped in managing and communicating prognostic uncertainty and making treatment recommendations when uncertainty was present [23, 45, 63, 76, 80, 85]. This culminated in doctors providing patients with what was viewed as sub-optimal and inconsistent information [76], requesting patients make treatment choices that risked contradicting their best interests and personal values [42, 57, 76, 81], or doctors avoiding goals of care discussions altogether [23, 49, 57, 63, 86].

Further communication skills training and support, for example, via mentoring were welcomed by doctors [19, 63, 67, 72]. However, the effectiveness of training interventions has not been established [76]. Uncertainties remain as to where the focus and content should lie, what are the most effective ways of delivering training, the optimal intensity and frequency of training interventions, how to measure their effectiveness in the short and longer term and how to engage less motivated learners [45]. The role of interpersonal accuracy and behavioural adaptability is also unclear. This relates to a doctor's ability to recognise emotions and motivating factors or thoughts in patients (interpersonal accuracy), and adapt their communication style accordingly (behavioural adaptability) [47]. In one study, female doctors had positive correlations between interpersonal accuracy and verbal and non-verbal behavioural adaptability which translated to more positive patient consultation outcomes (for non-verbal adaptability) [50]. However, for male doctors, better interpersonal accuracy was linked to less non-verbal adaptability for unclear reasons [50].

## B) Other external threats to trust

Pre-conceived views by patients and their families influenced trust in their treating doctor. Negative prior experiences with healthcare and information retrieved from the internet and other public sources of information contributed to a lower level of trust by the patient and family [60]. This was more prominent among those with relatively low health literacy, those who were less well educated, unemployed, with no medical insurance (in a fully privatised healthcare system), or those who were homeless [51]. Mistrust was also threatened when patients felt stigmatised because of their medical conditions such as human immunodeficiency virus and Acquired Immune deficiency syndrome (HIV/AIDS), psychiatric illness, including anorexia, bulimia, and substance abuse, sickle cell anaemia and other physical disabilities [51].

Cultural differences between doctors and patients and or families combined with a low level of cultural literacy (understanding) amongst doctors were detrimental to a trusting relationship [46]. Cultural differences might relate to "do not attempt cardiopulmonary resuscitation" (DNACPR) orders, the taboo around death and how death is perceived and communicated [46, 82]. However, families of culturally diverse backgrounds lacked awareness of how to communicate their cultural needs which sometimes resulted in conflict [49] or being stereotyped based on their religion or culture when their values were not always completely aligned [46].

Power differentials between doctors and patients and their families influenced trust [23, 63, 68, 80]. Furthermore, a disproportionate level of delegated power from doctor to patient and their family was considered a risk to the ongoing doctor-patient and family relationship, particularly when there were discordant views between both parties [23, 63, 68, 80].

## C) The Impact of Trust and Mistrust

Mutual trust between doctors and patients and their families facilitated improved understanding and positively enhanced ongoing relationships and rapport [60, 73]. Conversely, a lack of trust or misconception about treatments translated to feelings of abandonment, neglect or inferior care by the patient and their family [60, 63]. This sometimes resulted in more invasive care that was not always perceived to be appropriate but thought necessary by the doctor to avoid further conflict [60]. Frequent end-of-life communication and conflict resolution [63] had the potential to cause a high emotional burden for the doctor, avoidance of further engagement between doctor and patient in goals of care planning, or a breakdown in mutual trust between the two respective parties [16, 23, 51, 86].

### CMO Five: Doctors that are better able to identify patients who are most likely to benefit from goals of care conversations in acute severe illness will prioritise those most in need of such conversations, ensuring these conversations are initiated

Conversations were delayed due to a lack of precipitating events that prompted doctors to have goals of care conversations [23]. In hospital, focus was often placed on physiological parameters without recognition of the dying process [86]. Moreover, some patients had unpredictable disease trajectories which made it difficult to judge the best time to have conversations [79]. Doctors were sometimes prompted by nurses to hold conversations when the nurse perceived a patient was deteriorating and felt a discussion was needed, or when assertive family members felt uninformed and desired further information [22, 85].

In quality improvement studies, electronic health alerts combined with coaching did not increase the proportion of documented goals of care conversations [66, 72]. Intervention bundles used to facilitate conversations increased the frequency of discussions but did not improve the quality of information discussed [63, 72]. It is unclear whether this was due to the direct impact of the bundle itself or whether the bundle prompted a culture shift that indirectly improved the frequency of discussions [63, 72].

### CMO Six: Doctors who see the benefits and value of holding goals of care conversations in acute severe illness are more likely to be motivated and incentivised to hold such conversations which increases the number of conversations that are initiated

Preventing non-beneficial treatment during acute hospitalisations was a motivator for doctors to hold goals of care discussions [67]. Doctors who also witnessed other benefits of complex interventions designed to promote goals of care conversations, for example, the Serious Illness Conversations Programme (SICP) were incentivised to engage. Perceived benefits included the ability to provide better patient care, giving the platform for patients to "open up" and adding to their skill mix [64]. Doctors also reported being more satisfied with their work due to being able to connect with the patient and family at a deeper level and reducing moral distress [64]. However, goals of care conversations were also perceived as a "tick box" exercise [82]. Moreover, it was not always obvious whose responsibility and role it was to hold or initiate them [42]. The path of least resistance was considered to be a disincentive to engage in early goals of care conversations. Specifically, it was considered to be easier to withdraw life-sustaining treatments once the patient had experienced such treatment and expressed that they did not want it to continue as opposed to deciding to withhold life-sustaining treatment, including accompanying patient and family discussions, from the outset [16].

### CMO Seven: Healthcare organisational "buy-in" and a better understanding of organisational related facilitators and barriers to conducting goals of care conversations will allow the necessary organisation changes to be made that promote and facilitate these conversations between doctors and patients in acute severe illness

Organisational policies that promoted shared decision-making and the presence of prior positive patient and or family experiences enhanced patient trust and their willingness to engage in goals of care conversations. Organisational barriers to goals of care conversations included the perceived lack of an appropriate location in a busy clinical environment to hold sensitive

discussions [23, 80] and lack of doctor time due to other competing interests [16, 23, 58, 70, 80, 82, 88]. This was further compounded by the lack of administrative support to arrange and document the conversation [80], the lack of continuity of care or when the informational needs of the patient or family were high or discordant views were present between doctor and patient and or family [16, 23, 80, 82, 88]. Mistrust in the organisation from the patient and family perspective based on prior negative experiences translated to concerns about inferior ward treatment for DNACPR decisions [70].

Despite their willingness to be more involved, nurses often felt underutilised and undervalued in goals of care conversations. They reported their value included merely acting as an intermediary role between doctors and patients and their families. For example, priming patients and their families for these discussions [16, 63], facilitating information exchange between doctors, patients and their families [16], playing a supportive follow-up role in situations when doctors delivered bad news and reinforcing information [63]. However, they felt they could improve continuity of care if organisational work challenges allowed them to do so. In the analysis of the SICP, having a unit champion to undertake the administrative roles relating to goals of care conversations was well received and had the potential to facilitate conversations further [64]. Importantly, none of the included studies explored the roles of other allied healthcare professionals, for example, physiotherapists, clinical psychologists and social workers. Aside from staffing, other themes were suggestive of institutions needing to be culturally aware and inclusive of the needs of patients from ethnically diverse and minoritised communities to maintain trust [51, 52].

## Discussion

We present the first realist review to develop and refine an initial theory to explain goals of care communication between doctors and patients with severe acute illness in hospital settings. Through examination of our seven CMOs, we identify that patient-centred care in acute illness in hospitals requires shared understanding between doctor and patient about treatment goals and priorities. Achieving this requires a universal yet nuanced understanding of the multiple facilitators, barriers and complexities involved across each of the stakeholders.

Whilst quality improvement initiatives have attempted to increase patient and family engagement in goals of care planning, for example, the AMBER Care bundle and ReSPECT [48, 91, 92] they have struggled to gain widespread adoption [22] or to become routinised into mainstream practice [93]. Follow-up evaluation studies highlight a mismatch in communication and understanding between doctor and patient [64] and difficulties in identifying the most suitable patients to hold goals of care [93]conversations with at the most appropriate times. Although these studies partially explain the reasons for the low uptake of these initiatives in clinical practice, which include, for example, the absence of a champion to consistently support the delivery of interventions with fidelity or the absence of adequate training [93], they nevertheless, overlook other important factors. The use of realist methodology has allowed us to explore this further. This approach offers advantages to a traditional systematic review because instead of aiming to address a solitary research question, this approach incorporates the most relevant factors that may influence goals of care discussions and how these factors may be inter-related with one another, culminating in a much broader understanding of what happens in real-life practice.

### Review and further refinement of CMO hypotheses

Further analysis and reasoning suggest that if we are to understand with more confidence what happens in real-life practice in goals of care planning in acute illness, considering each

CMO separately and in isolation is likely to be oversimplified. It is more appropriate to first consider the factors that influence to what extent goals of care conversations are initiated.

Here, the onus lies largely with the doctor and their motivation and willingness to engage. Our collated evidence suggests that this motivation requires "buy-in" from the doctor about the benefits of goals of care planning ("context" hypothesis 6) together with feeling appropriately trained, skilled and confident to hold such conversations ("context" hypothesis 4). Therefore hypotheses 4 and 6 are closely intertwined and share the same "mechanism", in this case, motivation, that translates to a new "outcome" of interest–initial engagement in goals of care conversations between doctor and patient. In addition, hypothesis 5 relates to a process or system, where patients most likely to benefit from goals of care conversations could be better identified ("context" hypothesis 5) to aid clinical prioritisation ("mechanism" hypothesis 5), in promoting goals of care conversations. However, existing evidence is lacking to support this theory at present.

The second consideration involves the nuances in how conversations are conducted once initial engagement has occurred. This involves a more complex relationship between the "context" and "mechanisms" of hypotheses 1 (information provision), 3 (mutual trust) and 4 (skills and confidence of doctors) than our initial theory suggested. Our proposed model (Fig 2) depicts how the "context" and "mechanisms" of these hypotheses may inter-relate and combine to achieve the final "outcome" of interest–a shared understanding between doctor and patient regarding treatment goals and priorities. In this model, it is of note that the skills, confidence and interpersonal relations of doctors ("context" hypothesis 4) play a dual role in conversation initiation and the conversation process after initiation via two separate "mechanisms" and is therefore a critical component of goals of care communication in this setting. Our data did not suggest any refinements needed for CMO2 (communication bias) or CMO 7 (organisational influence) which are both relevant.

## Exploring the evidence further

Initiation of goals of care conversations requires doctors to value the importance of and acknowledge the benefits of a patient-centred approach [64]. Decreasing the likelihood of non-beneficial interventions, providing better patient care and allowing more open communication with the patient and their relatives were reasons cited by doctors that motivated and incentivised them to engage with patients and families about goals of care [64]. However, despite this, time pressures from competing work and the lack of any validated tools to identify patients most likely to benefit from goals of care conversations at the most appropriate times present additional challenges [23, 79, 86]. Conversations held too early risk mistrust among patients, particularly when they perceive the information provided is not relevant to them at all, or just yet [22, 58]. Conversations held too late may be limited by high emotional states at crisis points that have the potential to contradict a patient's longer-term values and increase the risk of burdensome care [54, 65].

Evidence also supported doctors lacking confidence and competence in initiating and holding goals of care conversations often due to fear of conflict when discordant views were present or anxieties relating to discussing end-of-life issues [16, 23, 42, 45, 49, 57, 63, 76, 79–81, 84–86]. Communication skills training was frequently welcomed by doctors and studies show a range of training formats. However, evidence shows the effectiveness of any communication training is generally lacking [63, 72, 76, 83, 94, 95]

A key aim is for doctors to harbour a positive relationship with their patients and families and foster trust. Open and honest communication (in a clear way that the patient can understand), showing competence and confidence, being personable, and "warm" and continuity of

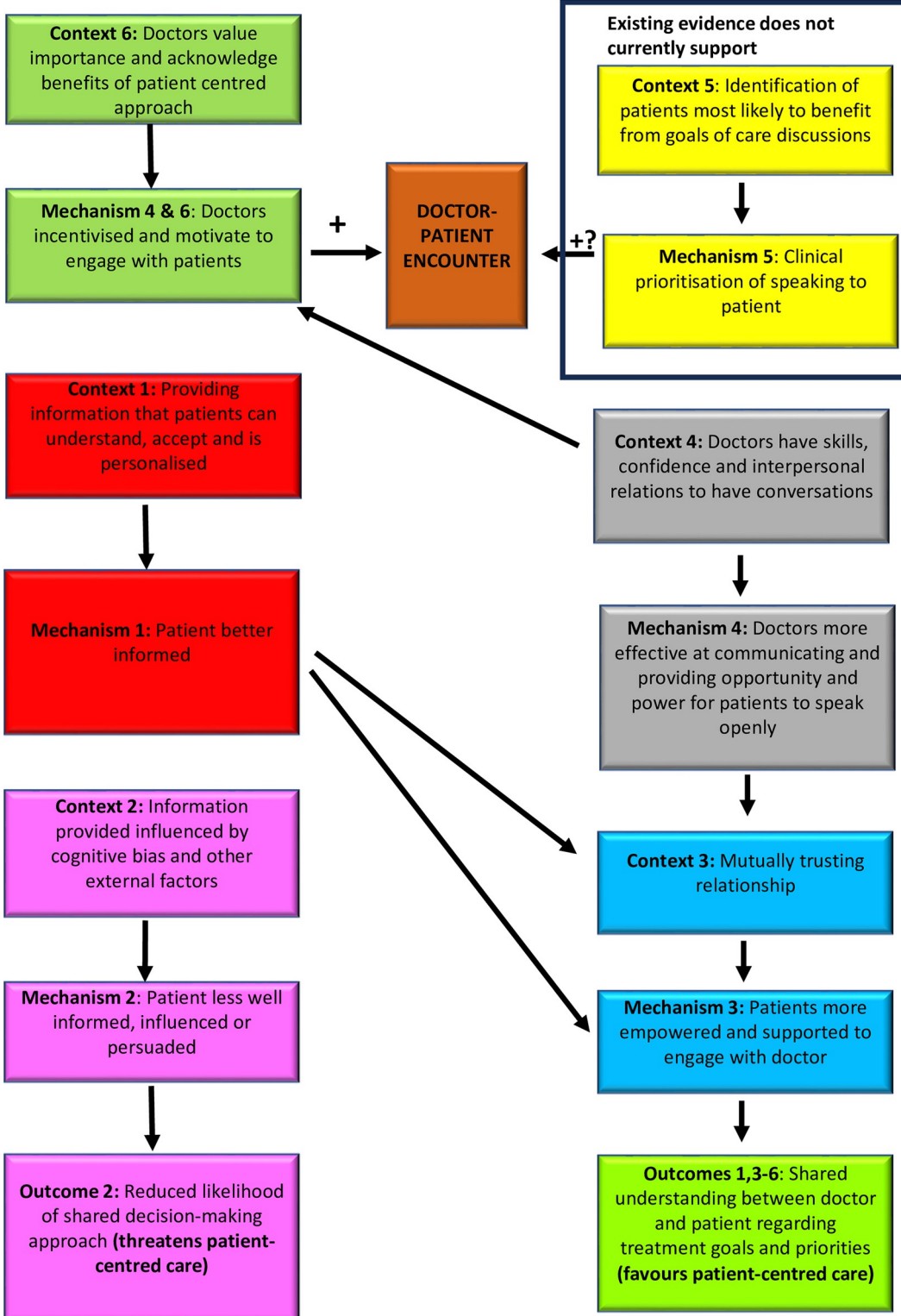

**Fig 2. Model proposing how CMO hypotheses may be inter-related.**

care were key factors described by patients/families to gain a trusting relationship with their doctor [58, 60, 61, 73]. However, information provided by doctors was frequently inconsistent [45, 84], biased [42, 54, 55, 58, 59, 69, 79], not personalised [42, 43, 45, 56, 58, 77, 82] and often poorly or misunderstood [42, 45, 58, 73]. These can negatively impact trust leading to discordant views, all of which have the potential to result in higher burdensome non-beneficial treatment [60]. Low cultural literacy and lack of cultural awareness amongst doctors risk them failing to adapt conversations to the needs of patients from minority ethnic communities. This can be of further detriment to trusting relationships between doctor, patient and family [46, 80].

## Proposals for further research, training and role allocation

Future work should focus on ways of ensuring that doctors are aware of and initiate and deliver goals of care conversations aligned to factors highlighted in this realist review. This could be accompanied by a realist evaluation [96–98] of training in this area with emphasis on how doctors are currently trained and how existing training could be modified to ensure doctors have the appropriate skills and confidence to effectively identify and communicate with patients and their families about goals of care in acute illness. Moreover, the development of validated training outcomes and patient experience measures would pave the way for the development of effective training methods and approaches [94, 95] Aside from doctor training, the role of nurses and other allied healthcare professionals requires further exploration. Our findings identify that nurses are currently underutilised and undervalued in goals of care conversations [16, 63]. Their contribution may be limited because there is no defined formal nursing role. However, nurses have considerable potential to facilitate communication between doctors and patients by enhancing mutual understanding, communicating patient concerns to doctors, and providing emotional support to patients and families. These have the potential to foster better relationships, increase the level of mutual trust and promote more timely conversations [16, 63, 78]. This is particularly relevant in a busy healthcare system where doctors are often time-pressured to have two-way communication between themselves and patients and or families. Although this review did not highlight any role for other allied healthcare professionals, for example, but not exclusive to physiotherapists, clinical psychologists, speech and language therapists and social workers; this may well be worth exploring further in future studies. Each allied health professional may offer a different perspective and skill mix, which may be of benefit to doctors, patients and their families in the context of goals of care discussions. Furthermore, this may also pave the way for multi-professional training which has been shown to improve confidence, knowledge and skills amongst healthcare professionals who underwent a training workshop in "difficult conversations" [95].

There is a need to further understand organisational and healthcare systems in addition to wider societal influences on the doctor-patient interaction. Finally, there is a need to explore the nuance of goals of care discussions. It is here that health professionals for example doctors may use a variety of 'voices': the 'doctor voice' to ask specific questions; the 'educator voice' to share information and help patients understand their illness, situation and treatment, and the 'fellow human voice' to convey empathy. By showing empathy through comments for example *"I understand"* or *"That must be really tough"*, health professionals share a 'fellow human voice' encouraging patients to discuss goals of care [33].

## Strengths and limitations of this review

A strength of this realist review is both its explanatory and theoretical nature- to understand the complex mechanisms underlying goals of care discussions with patients with severe acute

illness in hospital settings. This style of synthesis shifts the focus from specific interventions and services to broader underlying mechanisms or principles. Second, we actively incorporated key stakeholders' views alongside published literature to refine the review focus to areas considered most pertinent to clinical practice. Twenty-seven of the studies included within the review were qualitative and five were mixed method, a strength being that studies of this type permitted salient contexts, mechanisms and outcomes to be understood in detail, particularly where 'thick' accounts were evident.

However, the findings and recommendations of this review would not be complete without reference to the limitations of this work. Whilst a high number of studies in this review were qualitatively orientated which allow for a deeper understanding of the key concepts, few studies made use of methodological approaches that permit wider generalisation from their findings. Second, most of the identified studies were from Global North and Westernised countries (Australia, Canada, USA, UK or other European locations) and may not translate to non-westernised societies. Third, the influence of cultural values was not explored in detail nor was any detailed analysis of the family influence and patient-family relations [99]. This might include any possible impact of family persuasion or dominance in conversations between patients and doctors and discordant views between patients and families. Fourth, many included studies were classified as being 'weak' to 'moderate' concerning their methodological design and therefore subject to potential bias (Table 3). Fifth, this review did not explore how shared understanding between patient and doctor relates to the type of patient-doctor relationship. This relates to how decisions are made once there is mutual understanding between patient and doctor of treatment goals and priorities and incorporates a spectrum of decision-making from a paternalistic approach through to pure shared decision-making [100]. This is regarded as a separate entity and warrants a separate study. Sixth, multiple 'wrap-around' preconditions that underpin successful shared decision-making may also be present. This necessitates a socioecological lens [101] in which a whole systems strategic approach acknowledges multiple, interconnected elements potentially exist and reside at different societal and organisational levels of influence (microsystem (person, needs and characteristics), chronosystem (dynamic influences of time), mesosystem (interactions with family/ health professionals), exosystem (healthcare services/systems) and the macrosystem (societal influences). All may be necessary to consider before, during and after implementing shared decision-making conversations and warrant further exploration.

## Conclusion

This realist review highlights the factors that contribute to a shared understanding of treatment goals and priorities between patients living with acute life-threatening illnesses and healthcare professionals caring for them. Moreover, it examines the respective roles of patients, their families, and healthcare professionals, and the ways they inter-relate with each other throughout this process, which at times can be highly nuanced. The challenge now is to operationalise the ways this information provides mutual benefit to patients, their families and those who care for them, whilst acknowledging being flexible to the continually changing landscape of healthcare and wider society. Based on the findings from this review, we suggest that local educational hubs are organised, and geared towards continual professional development and learning to capacity-build healthcare professionals' skills and competencies when undertaking goals of care discussions. This could be achieved using repeated 'plan-do-check-act' (PDCA) cycle loops [102], either making use of role-play in a simulated environment or real-life, real-time settings, acquiring constructive feedback from colleagues and where appropriate from patients and their families. We also believe the themes from this review should serve to guide

the planning, execution and resultant learning from goals of care discussions, thereby continually modifying and developing clinical practice and experience. Consistent exposure and engagement in PDCA-specific goals of care discussions may improve competence and confidence over time, one of the barriers we identified in this review.

It is also vital to recognise the context of how these conversations are enacted. Recognition must also be placed on prior understanding of the most appropriate type of encounter between health care professionals and patients necessary for each clinical context [100]. For example, this may include a more paternalistic approach for situations where there is little or no chance of recovery or a more shared decision-making approach where situations of clinical uncertainty are present [103] and where a greater need for patient engagement is possible.

On a wider scale, the key is to provide value to doctors, patients and healthcare planners. Objective outcome measures that demonstrate effective communication in this field need to be defined and developed for each key stakeholder. From a healthcare professional and patient perspective, the focus may lie on improving the healthcare experience, whilst healthcare planners may also be interested in how more effective goals of care communication may positively influence other value-based metrics, for example, the use of scarce healthcare resources and ensuring they are equitably accessed by those who stand to benefit from them. This requires collaboration between all parties including service users to consider the most important outcome measures which could then be developed and validated using more traditional research methods. The combination of these shorter and longer-term strategies provides a foundation for further development towards optimal engagement and communication relating to goals of care.

## Supporting information

**S1 Checklist.**
(DOC)

**S2 Checklist.**
(DOCX)

**S1 Appendix. Search strategy.**
(DOCX)

**S2 Appendix. Evidence quality appraisal.**
(DOCX)

**S3 Appendix. Individual article relevance to CMOs.**
(DOCX)

**S4 Appendix. International prospective register of systematic reviews (PROSPERO) summary.**
(PDF)

## Author Contributions

**Conceptualization:** Jamie Gross, Jonathan Koffman.

**Data curation:** Jamie Gross, Jonathan Koffman.

**Formal analysis:** Jamie Gross, Jonathan Koffman.

**Investigation:** Jamie Gross, Jonathan Koffman.

**Methodology:** Jamie Gross, Jonathan Koffman.

**Project administration:** Jamie Gross, Jonathan Koffman.

**Supervision:** Jamie Gross, Jonathan Koffman.

**Validation:** Jamie Gross, Jonathan Koffman.

**Visualization:** Jamie Gross, Jonathan Koffman.

**Writing – original draft:** Jamie Gross.

**Writing – review & editing:** Jamie Gross, Jonathan Koffman.

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
