## [Decision Letter · Decision Letter 0]

20 Dec 2023

PONE-D-23-29556Examining how goals of care communication are conducted between doctors and patients with severe acute illness in hospital settings: A realist systematic reviewPLOS ONE

Dear Dr. Gross,

Thank you for submitting your manuscript to PLOS ONE. After careful consideration, we feel that it has merit but does not fully meet PLOS ONE’s publication criteria as it currently stands. Therefore, we invite you to submit a revised version of the manuscript that addresses the points raised during the review process.

We look forward to receiving your revised manuscript.

Kind regards,

Simon White

Academic Editor

PLOS ONE

Journal Requirements:

Did you know that depositing data in a repository is associated with up to a 25% citation advantage (https://doi.org/10.1371/journal.pone.0230416)? If you’ve not already done so, consider depositing your raw data in a repository to ensure your work is read, appreciated and cited by the largest possible audience. You’ll also earn an Accessible Data icon on your published paper if you deposit your data in any participating repository (https://plos.org/open-science/open-data/#accessible-data).

**Additional Editor Comments:**

This paper seems to have split the reviewers and I think this largely relates to the process of undertaking a realist synthesis and the role of the stakeholder consultation within it. The methods and results of the stakeholder consultation appear to be both presented in the methods section and this makes it difficult to follow the outcomes at each stage of the process. In addition, the refinements and revisions made to the CMO configurations as a result of the data extraction and synthesis process are not clear – a summary of these and presentation of the finalised wording of CMOs in the results section would benefit the paper. If this is what is shown in table 2, this should be explained, as presently it appears that this is the outcome of the stakeholder consultation exercise, and it is not clear why this table is in the methods rather than in the results section. As a minor point, in table 2 – CMO 6 should read “…then they will be incentivised…”. Please address these points, in addition to those made by the reviewers below.

Reviewers' comments:

Reviewer's Responses to Questions

**Comments to the Author**

1. Is the manuscript technically sound, and do the data support the conclusions?

Reviewer #1: Partly

Reviewer #2: Partly

Reviewer #3: Yes

2. Has the statistical analysis been performed appropriately and rigorously? 

Reviewer #1: No

Reviewer #2: Yes

Reviewer #3: N/A

3. Have the authors made all data underlying the findings in their manuscript fully available?

Reviewer #1: Yes

Reviewer #2: Yes

Reviewer #3: Yes

4. Is the manuscript presented in an intelligible fashion and written in standard English?

Reviewer #1: Yes

Reviewer #2: Yes

Reviewer #3: Yes

5. Review Comments to the Author

Reviewer #1: "This article effectively examines articles related to the enhancement of patient satisfaction through goals of care decision-making. However, in the analysis, it is crucial to perform a quality assessment, and models such as JBI or other applicable frameworks can be employed for this purpose."

Reviewer #2: Thank you for inviting me to review the manuscript entitled “Examining how goals of care communication are conducted between doctors and patients with severe acute illness in hospital settings: A realist synthesis review”. The manuscript is of an important topic and well-written, however, I do find the methods would benefit from further clarity before publication. Please find my comments below:

Line 51: What do the authors mean by ‘future directions’?

Line 56: Is there a reference regarding patient satisfaction?

Line 98: What do you mean by a realist systematic review? It seems like the authors are combining the methodology of a realist synthesis of the literature and a systematic review, more clarity is needed.

Line 124: You describe the involvement of key stakeholders within the realist synthesis. I think this would benefit from being within its own section. How were these ‘key’ stakeholders identified? How did you present the CMOs to the stakeholders? What was the impact of these discussions?

Line 130: ‘Each consultation’ sounds clinical – should it be ‘each discussion’?

Line 132: When there was ‘ambiguity’ or ‘doubt’, why was agreement only sought between JG and JK?

Generally, do the authors have a description of their initial theory? I find that this reads as a systematic review, not a realist synthesis.

Line 218 (and the rest of the results): Generally realist results are presented in CMO configurations. I cannot follow what was a context-mechanism or outcome through-out the whole results section. I suggest re-structuring this section into CMO configurations or going through the section highlighting what is a context, mechanism and outcome.

The results section would also benefit from a diagram which shows how the CMOs relate to each other.

The discussion is well-written, but could benefit from sub-headings to help the narrative.

Reviewer #3: This was a very well planned and undertaken study. The realist systematic review was well aligned to the research outcomes. The introduction set the scene well, justifying the need for the review and how the review will help understand the problems currently faced in goals of care decisions making between doctors and patients. The introduction may have been extended to highlight what the expectation of good decision making may look like.

Methods are well structures, follow guidance expected for a realist review and has been registered with PROSPERO. Appended information show how the study has been aligned with PRISMA and RAMESES checklists as would be expected in this review. There was a very nice description of the context-mechanism-outcome configuration and the development of the initial programme theory was outlined well along with the scoping literature search.

A comprehensive list of search strategy was included in appendix 1. It would have been nice in the methods to outline geographic restrictions (if any) as it is not mentioned until later in the results/discussion sections. The extracting data section may have been further strengthened with more detail on the strategies employed to refine and revise the CMOs, with more detail given to how the notes/schematic worked in providing the audit trail of decisions made.

The results section was well laid out, with clear relation to the initial programme theory and quotes were used well to highlight themes/subthemes within each CMO.

A little more reflection/alignment of the discussion text to figure 2 may have helped orientate the reader a little more.

Is there any other data beyond that presented in Table 4 that should be shared? For example similar quotes to those selected that could be appended?

6. PLOS authors have the option to publish the peer review history of their article (what does this mean?). If published, this will include your full peer review and any attached files.

Reviewer #1: No

Reviewer #2: No

Reviewer #3: No

---

## [Author Response · Author response to Decision Letter 0]

30 Jan 2024

Additional Editor Comments:

This paper seems to have split the reviewers and I think this largely relates to the process of undertaking a realist synthesis and the role of the stakeholder consultation within it. The methods and results of the stakeholder consultation appear to be both presented in the methods section and this makes it difficult to follow the outcomes at each stage of the process. In addition, the refinements and revisions made to the CMO configurations as a result of the data extraction and synthesis process are not clear – a summary of these and presentation of the finalised wording of CMOs in the results section would benefit the paper. If this is what is shown in table 2, this should be explained, as presently it appears that this is the outcome of the stakeholder consultation exercise, and it is not clear why this table is in the methods rather than in the results section. As a minor point, in table 2 – CMO 6 should read “…then they will be incentivised…”. Please address these points, in addition to those made by the reviewers below.

Thank you for your comments. We feel that it is appropriate to keep the methods and results of the stakeholder consultation discretely in the methods section. This is because the stakeholder forms part of the methodology for a realist synthesis as described in: 

Rycroft-Malone, J., McCormack, B., Hutchinson, A.M et al. Realist synthesis: illustrating the method for implementation research. Implementation Sci 7, 33 (2012). https://doi.org/10.1186/1748-5908-7-33

and reported in other realist reviews similar to this manuscript for example: 

Cottrell L, Economos G, Evans C, Silber E, Burman R, et al. (2020) A realist review of advance care planning for people with multiple sclerosis and their families. PLOS ONE 15(10): e0240815.https://doi.org/10.1371/journal.pone.0240815

As with all realist reviews, the purpose of the stakeholder consultation is to optimise the strength of our hypotheses which then informed our search strategy of the literature. We feel the results section should be reserved for article findings following that literature search. This would be consistent with a traditional systematic review.

We agree with the points you raise about the CMOs. We have now provided the finalised wording of the CMOs as suggested in table 2 in the methods section (p8-9) and the subtitle headings for each CMO in the results section. We have also addressed the refinements and revisions made to the CMO configurations following data extraction and synthesis. These are now contained as a separate section under the heading “review and further refinement of the CMO hypotheses” in the discussion section, specifically on pages 46-47 of the revised manuscript. In this section, Figure 2 is referred to and explained as suggested.

Thank you for highlighting the anomaly CMO 6. This has now been addressed on page 9 of the revised manuscript.

Reviewer #1: This article effectively examines articles related to the enhancement of patient satisfaction through goals of care decision-making. However, in the analysis, it is crucial to perform a quality assessment, and models such as JBI or other applicable frameworks can be employed for this purpose."

We thank Reviewer 1 for their comments. A quality assessment was indeed performed using established appraisal checklists, which are described on P10 lines 195-199 of the revised manuscript. However, we concede that this is not transparent in the original submission. We have therefore updated the manuscript which now explains how papers were given ‘low’, ‘medium’ and ‘high’ quality ratings. This can be found under the “Rigour Screening” subheading in the methods section on pages 10-11 of the revised manuscript. For full transparency, we have now included a quality assessment of every included paper according to the appropriate appraisal checklist. This can be viewed as a supplement file in Appendix 2.

Reviewer #2: Thank you for inviting me to review the manuscript entitled “Examining how goals of care communication are conducted between doctors and patients with severe acute illness in hospital settings: A realist synthesis review”. The manuscript is of an important topic and well-written, however, I do find the methods would benefit from further clarity before publication.

We thank Reviewer 2 for their comment. We also agree that this is an important topic to examine and have taken steps to enhance the clarity of the methods section . 

Please find my comments below:

Line 51: What do the authors mean by ‘future directions’?

We agree that the meaning of “future directions” may not be clear to all readers. We have now modified this sentence on page 3 lines 50-51 of the revised manuscript which now reads “Implications for policy, research and clinical practice including further training and skills development for doctors are discussed”. We believe this is now more specific

Line 56: Is there a reference regarding patient satisfaction? 

We have now included relevant references associated with shared decision-making and improved patient satisfaction.

Line 98: What do you mean by a realist systematic review? It seems like the authors are combining the methodology of a realist synthesis of the literature and a systematic review, more clarity is needed.

A realist review is a now well-established approach to undertaking a systematic review. This is well described in the following references, which are cited in our manuscript: 

Pawson R, Greenhalgh T, Harvey G, Walshe K. Realist review--a new method of systematic review designed for complex policy interventions. J Health Serv Res Policy. 2005 Jul;10 Suppl 1:21-34. doi: 10.1258/1355819054308530. 

Rycroft-Malone, J., McCormack, B., Hutchinson, A.M. et al. Realist synthesis: illustrating the method for implementation research. Implementation Sci 7, 33 (2012). https://doi.org/10.1186/1748-5908-7-33

Because our methodology is consistent with that set out in these references, we do not feel that any modification is needed.

Line 124: You describe the involvement of key stakeholders within the realist synthesis. I think this would benefit from being within its own section. How were these ‘key’ stakeholders identified? How did you present the CMOs to the stakeholders? What was the impact of these discussions?

We agree that the manuscript would benefit from a more detailed explanation of stakeholder involvement. We now provide more detail on pages 7-8 of the revised manuscript which addresses all points raised.

Line 130: ‘Each consultation’ sounds clinical – should it be ‘each discussion’?

The term “consultation” [with stakeholders] is widely accepted as being appropriate in realist methodology. Please refer to 

Rycroft-Malone, J., McCormack, B., Hutchinson, A.M et al. Realist synthesis: illustrating the method for implementation research. Implementation Sci 7, 33 (2012). https://doi.org/10.1186/1748-5908-7-33) and has been reported in this manner in many previous realist review articles for example, R Hunter et al Realist review International Review of Sport and Exercise Psychology, 2022 15:1, 242-265, DOI: 10.1080/1750984X.2021.1969674 or Cottrell L, Economos G, Evans C, Silber E, Burman R, et al. (2020) A realist review of advance care planning for people with multiple sclerosis and their families. PLOS ONE 15(10): e0240815.https://doi.org/10.1371/journal.pone.0240815. 

Consequently, we do not believe a change is terminology is required.

Line 132: When there was ‘ambiguity’ or ‘doubt’, why was agreement only sought between JG and JK?

We thank Reviewer 2 for this helpful comment. For clarity throughout the entire process of undertaking the realist review, JG and JK had regular communication and discussions about theory and field notes were kept accordingly. However, we accept that the sentence you refer to may cause confusion and therefore have removed it. 

Generally, do the authors have a description of their initial theory? I find that this reads as a systematic review, not a realist synthesis.

We very respectfully take issue with Reviewer 2’s comment here. We have indeed provided a series of underlying programme theories and then interrogated existing evidence to find out where these theories are pertinent and productive to address the overarching review question. This primary research was then critically examined for its contribution to the developing theory. This corresponds exactly with the guidance provided by Pawson R: Evidence-based Policy. A Realist Perspective. 2006, London: Sage

Line 218 (and the rest of the results): Generally realist results are presented in CMO configurations. I cannot follow what was a context-mechanism or outcome through-out the whole results section. I suggest re-structuring this section into CMO configurations or going through the section highlighting what is a context, mechanism and outcome.

We agree with this very helpful observation. We have now included a more complete description for each CMO subheading which we believe will make the CMO configurations easier to follow.

The results section would also benefit from a diagram which shows how the CMOs relate to each other.

A figure illustrating how CMOs relate to each other was already included in the discussion of the original manuscript (figure 2). We believe that this should remain in the discussion. It is our view the results section should be focussed on reporting the key findings extracted from the relevant included papers to support each of the CMO hypotheses and the discussion is better suited to the interpretation of these findings. We feel that part of the interpretation of findings includes how the CMOs relate to each other, hence the need for the figure here. We have updated this section to include a more detailed explanation about how the CMOs relate to each other and importantly, how the CMO configurations have changed following this process. Moreover, we have also updated Figure 2 to support the text explaining this. 

The discussion is well written, but could benefit from sub-headings to help the narrative.

We agree with this helpful suggestion and have further subdivided the discussion into several sub-sections to assist the reader.

Reviewer #3: This was a very well planned and undertaken study. The realist systematic review was well aligned to the research outcomes. The introduction set the scene well, justifying the need for the review and how the review will help understand the problems currently faced in goals of care decisions making between doctors and patients. The introduction may have been extended to highlight what the expectation of good decision making may look like.

We thank Reviewer 3 for their helpful comments. On page 4 of the revised manuscript, we have extended the first paragraph which now includes an evidence-based definition of ‘goals of care’ and describes its important elements (lines 57-63). We believe this addresses Reviewer 3’s suggestion of highlighting what good decision-making corresponds to and sets the scene for the rest of the paper.

Methods are well structured, follow guidance expected for a realist review and has been registered with PROSPERO. Appended information show how the study has been aligned with PRISMA and RAMESES checklists as would be expected in this review. There was a very nice description of the context-mechanism-outcome configuration and the development of the initial programme theory was outlined well along with the scoping literature search.

We thank Reviewer 3 for sharing these comments

A comprehensive list of search strategy was included in appendix 1. It would have been nice in the methods to outline geographic restrictions (if any) as it is not mentioned until later in the results/discussion sections. 

We have added now added a sentence (lines 174-175) on page 9 of the revised manuscript) which states: “Although there were no geographic restrictions, papers were limited to those in written in English”

The extracting data section may have been further strengthened with more detail on the strategies employed to refine and revise the CMOs, with more detail given to how the notes/schematic worked in providing the audit trail of decisions made.

We thank Reviewer 3 for this helpful comment. For clarity throughout the entire process of undertaking the realist review, JG and JK had regular communication and discussions about theory development. Field notes were kept accordingly. We feel that this is explained adequately enough in the manuscript and therefore do not feel that any modification is needed.

The results section was well laid out, with clear relation to the initial programme theory and quotes were used well to highlight themes/subthemes within each CMO. A little more reflection/alignment of the discussion text to Figure 2 may have helped orientate the reader a little more.

On pages 46-47 of the revised manuscript, the text has now been modified to include a more detailed explanation in the text of how the CMOs relate to each other and has been modified as a result of this realist review process. Figure 2 has now been revised and is consistent with what has been written in the text.

Is there any other data beyond that presented in Table 4 that should be shared? For example similar quotes to those selected that could be appended?

This was already included as an appendix. In the revised submission, please refer to Appendix 3.

---

## [Editor Report · Decision Letter 1]

19 Feb 2024

Examining how goals of care communication are conducted between doctors and patients with severe acute illness in hospital settings: A realist systematic review

PONE-D-23-29556R1

Dear Dr. Gross,

We’re pleased to inform you that your manuscript has been judged scientifically suitable for publication and will be formally accepted for publication once it meets all outstanding technical requirements.

Kind regards,

Simon White

Academic Editor

PLOS ONE
---

## [Editor Report · Acceptance letter]

3 Mar 2024

PONE-D-23-29556R1 

PLOS ONE

Dear Dr. Gross, 

I'm pleased to inform you that your manuscript has been deemed suitable for publication in PLOS ONE. Congratulations! Your manuscript is now being handed over to our production team.

Kind regards, 

on behalf of

Dr. Simon White 

Academic Editor

PLOS ONE